# Loss of *Mir146b* with aging contributes to inflammation and mitochondrial dysfunction in thioglycollate-elicited peritoneal macrophages

Andrea Santeford[1], Aaron Y Lee[1†], Abdoulaye Sene[1‡], Lynn M Hassman[1], Alexey A Sergushichev[2§], Ekaterina Loginicheva[2], Maxim N Artyomov[2], Philip A Ruzycki[1], Rajendra S Apte[1,3,4*]

[1]Department of Ophthalmology and Visual Sciences, Washington University in St. Louis School of Medicine, St. Louis, United States; [2]Department of Pathology and Immunology, Washington University in St. Louis School of Medicine, St. Louis, United States; [3]Department of Medicine, Washington University in St. Louis School of Medicine, St. Louis, United States; [4]Department of Developmental Biology, Washington University in St. Louis School of Medicine, St. Louis, United States

**\*For correspondence:**
apte@wustl.edu

**Present address:** [†]Department of Ophthalmology, University of Washington, Seattle, United States; [‡]Kartos Therapeutics, Redwood, United States; [§]ITMO University, Computer Technologies Laboratory, Saint Petersburg, Russia

**Competing interests:** The authors declare that no competing interests exist.

**Abstract** Macrophages undergo programmatic changes with age, leading to altered cytokine polarization and immune dysfunction, shifting these critical immune cells from protective sentinels to disease promoters. The molecular mechanisms underlying macrophage inflammaging are poorly understood. Using an unbiased RNA sequencing (RNA-seq) approach, we identified *Mir146b* as a microRNA whose expression progressively and unidirectionally declined with age in thioglycollate-elicited murine macrophages. *Mir146b* deficiency led to altered macrophage cytokine expression and reduced mitochondrial metabolic activity, two hallmarks of cellular aging. Single-cell RNA-seq identified patterns of altered inflammation and interferon gamma signaling in *Mir146b*-deficient macrophages. Identification of *Mir146b* as a potential regulator of macrophage aging provides novel insights into immune dysfunction associated with aging.

## Introduction

Macrophages are innate immune cells that perform critical surveillance functions and phagocytose pathogens and cellular debris (*Mosser and Edwards, 2008*; *Wang et al., 2019*). Through surface-bound or secreted factors, they signal to other cells and regulate diverse processes such as angiogenesis, inflammation, and fibrosis. These global functional consequences of macrophage activity make these cells critical in regulating the molecular pathogenesis of cardiovascular disease, cancer, neurodegenerative disorders of the central nervous system, and blinding eye conditions (*Apte et al., 2006*; *Kelly et al., 2007*; *Nakao et al., 2005*; *Moore and Tabas, 2011*; *Mammana et al., 2018*). Macrophages undergo broad programmatic changes with aging that manifest as abnormal macrophage activation and polarization and are phenotypically labeled as immunosenescence or inflammaging (*Xia et al., 2016*; *Lin et al., 2018*; *Sene and Apte, 2014*). This age-associated shift in the macrophage phenotype from disease protective to disease promoting led us to hypothesize that alterations in the aged macrophage transcriptome may regulate this dysfunction.

Altered macrophage polarization and activation are associated with aging and drive molecular inflammation. Although this age-induced macrophage-mediated phenotype has been partially characterized, the current paradigm relies on an incomplete cytokine signature to determine whether

macrophages mitigate or promote disease with little information about the altered regulatory networks that inform downstream effector function of aging macrophages (*Locati et al., 2013*; *Nakamura et al., 2015*). Over the past decade, numerous studies have described how microRNAs (miRNAs), short non-coding RNAs ~22 nucleotides long, regulate gene expression by either transcript degradation or translational repression (*Bartel, 2009*). Multiple laboratories have demonstrated that miRNAs control critical processes in macrophages, including cholesterol efflux, lipid metabolism, and polarization (*Lin et al., 2018*; *Sene et al., 2013*; *Banerjee et al., 2013*; *Cai et al., 2012*). We hypothesized that expression of specific macrophage miRNAs is altered with organismal aging and may drive the inflammaging process. We tested this by examining changes in miRNA expression by RNA sequencing (RNA-seq). Using this unbiased approach, we identified a potentially high-value miRNA (*Mir146b-5p*) that inversely correlates with thioglycollate-elicited macrophage (TGEM) host age. Here we demonstrate that expression of *Mir146b* progressively declines in the aging TGEM and is associated with significant mitochondrial dysfunction and abnormal macrophage activation and polarization, recapitulating the inflammaging phenotype.

## Results

### TGEM purity is unaffected by host age

Although miRNAs have been implicated in regulating age-associated gene expression, their role in directing gene expression patterns and function in aging macrophages is unclear. We hypothesized that altered transcriptional regulation by miRNAs contributed to age-associated programmatic alterations in macrophages and sought to examine these changes in non-coding RNA expression on a genome-wide scale. Thioglycollate injection induces a sterile inflammatory response in the mouse peritoneum, which elicits infiltration of monocytes from the blood. Lavage of the peritoneum several days post-injection allows for the collection of the activated TGEM population, consisting of both resident and recruited macrophages (*Pavlou et al., 2017*; *Ghosn et al., 2010*; *Layoun et al., 2015*), with the potential to collect additional cell types that may be present in the peritoneum. Collected cells are gently centrifuged, resuspended in complete culture medium containing fetal bovine serum (FBS), and plated in tissue culture plastics. TGEMs adhere to tissue culture plastics, while other cells such as red blood cells or adipocytes do not. Cells that have not adhered to the tissue culture plates can be eliminated by gentle washes with Dulbecco's phosphate-buffered saline (DPBS) and the remaining adhered cells can be used for analysis. In order to ensure that this preparation method provided a highly pure TGEM population for our studies and to determine if the age of the mouse from which the TGEMs were harvested impacted the degree of macrophage purity, we harvested peritoneal lavage exudates from five young (3 months old) and five old (20 months old) female C57Bl/6 mice 4 days post-thioglycollate induction. Cells collected from individual mice were treated as independent biological samples. We plated samples as described above and allowed the macrophage population to adhere overnight to the culture dishes. The next day, non-adherent cells were washed away using DPBS and we gently lifted the remaining adherent cells into freshly prepared ice-cold fluorescence-activated cell sorting (FACS) buffer. Each sample was stained for macrophage markers CD11b and F4/80. Flow cytometry analysis showed that approximately 95% of all cells stained double-positive for both markers, confirming these cells as macrophages. Moreover, we confirmed that the age of the mouse from which the cells were harvested does not impact the purity of the TGEM population obtained through this procedure (*Figure 1—figure supplement 1*). We utilized this method of thioglycollate induction and cell adherence to collect TGEMs for the experiments throughout this study.

### Macrophage *Mir146b* expression levels decline during aging

Having established our method for obtaining TGEMs, we next performed RNA-seq analysis using small RNAs purified from TGEMs isolated from six time points (3 months, 6 months, 12 months, 18 months, 24 months, and 30 months) spanning the normal lifespan of C57Bl/6 mice to identify miRNAs whose expression changed with aging. Peritoneal exudates from 10 female thioglycollate-elicited mice were pooled to obtain one TGEM sample per time point. As miRNAs generally function to repress their targets, we were most interested in miRNAs whose expression decreased with organismal age, which would in turn lead to the accumulation of the miRNA's downstream target

genes that may contribute to age-associated inflammation and cellular dysfunction. Unsurprisingly, we found many miRNAs whose overall expression was altered with age (*Figure 1—figure supplement 2A*). We next validated the miRNAs whose expression seemed to decrease most gradually with increasing host age using quantitative polymerase chain reaction (qPCR). Each individual mouse used was treated as an independent biological replicate sample (n = 7–9 mice per age group). Each sample was run in duplicate for each probe set, and the average cycle threshold (Ct) value of the two technical replicates for each sample was used for computing the ΔΔCt and expression levels relative to our youngest (3 months old) time point. While our original RNA-seq dataset included mice as old as 30 months, the oldest mice available at the time of our validation were aged 20 months. Utilizing TGEMs from female mice at ages 3, 12, or 20 months, we observed no statistically significant differences (p<0.05) in expression levels of *Mir15a*, *Mir29a*, *Mir423*, *Mir146a*, or *Mir18a* with age. *Mir362* (data not shown) was also evaluated but was undetected for approximately half of all samples tested, independent of age, and was therefore excluded from analysis. Two miRNAs, *Mir146b* and *Mir22*, displayed significant decreases from 3 to 20 months (*Figure 1—figure supplement 2B–H*). Previous studies in monocytes/macrophages have demonstrated that *Mir22* is upregulated through PU.1 during hematopoetic differentiation and a loss of *Mir22* is associated with acute myeloid leukemia (*Shen et al., 2016*; *Jiang et al., 2016*). *Mir22* can act as either a tumor suppressor or an oncomiR, depending upon the context, and its expression has been shown to increase with aging in hearts (*Huang and Wang, 2018*; *Wang et al., 2017*; *Jazbutyte et al., 2013*). For *Mir146b*, prior works have demonstrated decreased expression in obesity and the progression and metastases of cancers including T-cell acute lymphoblastic leukemia and glioblastoma multiforme, and its expression has been noted to decrease with age in the lungs in human bronchial biopsies from healthy individuals (*Hulsmans et al., 2012*; *Correia et al., 2016*; *Li et al., 2013*; *Ong et al., 2019*). The effects of age-associated loss of neither *Mir22* nor *Mir146b* have been previously demonstrated in macrophages. As the normalized expression of *Mir146b* in our RNA-seq dataset was greater than 3.3-fold at 3 months compared to that of *Mir22* (*Figure 1A* and *Figure 1—figure supplement 2I*), indicating a potentially more biologically relevant target in this cell type, we chose to further investigate the role of *Mir146b* in TGEMs.

## *Mir146b* expression is decreased with aging in TGEMs and BMDMs

Having noted that *Mir146b* expression decreased in TGEMs from 3 to 20 months, we next asked whether this decrease continued through 30 months of age (in accordance with our RNA-seq time points) and whether this trend also occurs in other macrophages, such as bone marrow-derived macrophages (BMDMs), which represent a more naive state compared to elicited TGEMs. We indeed observed a decrease of more than twofold in *Mir146b* gene expression in TGEMs isolated from 3-month-old vs 30-month-old female mice (*Figure 1B*), which was similar to the decrease noted for the same time points from our RNA-seq experiment. Levels of *Mir146b* also declined with age in BMDMs (*Figure 1C*); however, *Mir146b* relative expression in unstimulated BMDMs from individual 3-month-old female mice was only ~20% of that in TGEMs (*Figure 1D*). To further validate these findings, we used the hybridization probe-based Affymetrix QuantiGene 2.0 miRNA plate assay. We isolated total RNA from female TGEMs at 3 or 30 months of age and applied 250 ng of total RNA per well containing hybridization probe sets targeting either mature *Mir146b* or housekeeping gene *U6* for overnight incubation. Two technical replicates per probe set were run for each biological sample/mouse. The following day, the signal was amplified and detected by using a chemiluminescent substrate according to the manufacturer's instructions. Here we further confirmed a comparable decrease in *Mir146b* relative expression levels from 3 to 30 months in TGEMs (*Figure 1E*). Expression was below the limit of detection in naive BMDMs (data not shown) from both young and old female mice. Given the more robust expression of *Mir146b* in TGEMs, detectable without added stimulation, we continued our studies utilizing these cells exclusively.

## Macrophage *Mir146b* regulates cytokine expression

Small RNAs such as miRNAs primarily regulate gene silencing within cells by binding to canonical target seed sequences in the 3' untranslated regions (UTRs) of protein coding genes to initiate a process of post-transcriptional degradation (*Bartel, 2009*). Age causes poorly understood programmatic changes within the macrophage that lead to a shift in polarization that promotes dysfunctional

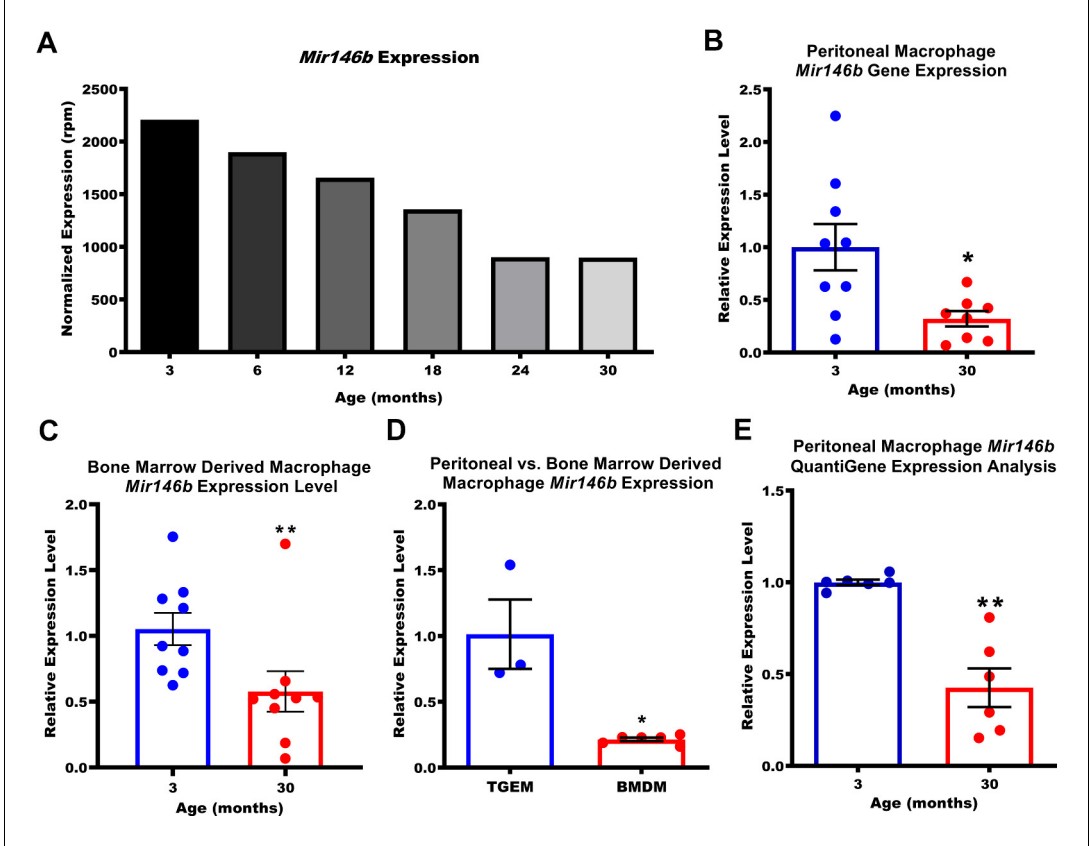

**Figure 1.** Expression of macrophage *Mir146b* declines with aging. (**A**) Small RNA transcriptomic profiling of thioglycollate-elicited macrophages (TGEMs) across the mouse virtual lifespan identified *Mir146b* as a microRNA (miRNA) whose transcription levels progressively and unidirectionally decreased with host age (3- to 30-month-old C57Bl/6 females; TGEMs were pooled from n = 10 mice into a single sample per time point) (2207.22 reads per million [rpm] at 3 months vs 897.94 rpm at 30 months). (**B**) Representative quantitative polymerase chain reaction (qPCR) gene expression analysis of *Mir146b* from TGEMs from C57Bl/6 female mice (n = 9 vs n = 8 mice [biological replicates] from three independent experiments; *p = 0.0206) and (**C**) unstimulated bone marrow-derived macrophages (BMDMs) (n = 9 vs n = 9 mice [biological replicates] from three independent experiments; **p = 0.0040) from young (3 months) or old (30 months) C57Bl/6 female mice. (**D**) Relative qPCR expression levels of *Mir146b* in TGEMs compared to BMDMs (n = 3 vs n = 6 mice [biological replicates]; *p = 0.0238). (**E**) Relative *Mir146b* expression measured by Affymetrix QuantiGene 2.0 miRNA assay in TGEMs from 3-month- or 30-month-old female C57Bl/6 mice. Graph showing data from three independent experiments of n = 2 mice (biological replicates) per group (n = 6 vs n = 6; **p = 0.0022). Data for (**B-E**) are represented as mean ± SEM. Each point represents the mean value from two technical replicates per mouse. Mann-Whitney U-test was used for comparison between two groups.

The online version of this article includes the following source data and figure supplement(s) for figure 1:

**Source data 1.** Flow cytometry density plots from TGEMs isolated from (n = 5) 3-month (bottom row)- and (n = 5) 20-month-old (top row) mice and stained with CD11b eFluor 450 and F4/80 Allophycocyanin (APC) along with numerical values for each mouse indicating the percentage of cells stained double-positive for both markers (CD11b+/F4/80+).

**Source data 2.** miRNA expression values in reads per million from small RNA-seq of female TGEMs at 3, 6, 12, 18, 24, and 30 months of age (related to *Figure 1A* and *Figure 1—figure supplement 2A and I*).

**Source data 3.** Relative miRNA expression values from qPCR of TGEMs from 3-month (n = 9 mice)-, 12-month (n = 8 mice)-, and 20-month (n = 9 mice)-old female mice.

**Source data 4.** Relative *Mir146b* miRNA expression values used for the graph from qPCR of TGEMs from n = 9 young (3 months) or n = 8 old (30 months) female mice (related to *Figure 1B*).

**Source data 5.** Relative *Mir146b* miRNA expression values used for the graph from qPCR of BMDMs from n = 9 young (3 months) or n = 9 old (30 months) female mice (related to *Figure 1C* ).

**Source data 6.** Relative *Mir146b* miRNA expression values used for graphs from qPCR of young female TGEMs (n = 3 mice) vs BMDMs (n = 6 mice) (related to *Figure 1D*).

**Source data 7.** Relative *Mir146b* miRNA expression values used for graphs from Quantigene 2.0 assay of TGEMs from n = 6 young (3 months) or n = 6 old (30 months) female mice (related to *Figure 1E*).

**Figure supplement 1.** Thioglycollate-elicited macrophage purity is consistent between young and old mice.

**Figure supplement 2.** *Mir146b* and *Mir22* are downregulated with age in murine TGEMs.

immunity and disease (*Wang et al., 2019*). RNA-seq and subsequent qPCR validation demonstrated that *Mir146b* showed a consistent unidirectional alteration in expression pattern over the lifetime of a mouse. As such, we hypothesized that *Mir146b* target genes may either directly or indirectly regulate macrophage polarization and aging phenotype. High levels of *Mir146b* may promote a 'young' phenotype, and the progressive decline in its expression seen with increasing age likely leads to the alternative 'old' macrophage phenotype. To assess this hypothesis, we created macrophages (from young 3-month-old C57Bl/6 female mice) with transient *Mir146b* knockdown through fast-forward transfection with *Mir146b*-specific antagomir inhibitors at a final concentration of 25 nM. We measured the transfection efficiency (in separate wells) using a fluorescently tagged sham inhibitor of similar nucleotide length and noted a transfection efficiency of approximately 80%, with less than 2% cell death assessed by terminal deoxynucleotidyl transferase dUTP nick end labeling (TUNEL) (*Figure 2A* and data not shown) 24 hr post-transfection. This transfection protocol resulted in >50% reduction in macrophage *Mir146b* miRNA expression levels when measured by qPCR (*Figure 2B*). Knockdown of *Mir146b* for 72 hr in TGEMs resulted in trends of decreased gene expression of multiple cytokines and regulatory markers including *Nos2*, *Mmp9*, *Il1b*, *Il6*, *Arg1*, and *Cd163* along with a trend toward increased *Il10* (*Figure 2C*) when compared to the relative expression in sham-transfected control TGEMs. These results implicate *Mir146b* in regulating genes associated with macrophage polarization and inflammation, phenocopying aging macrophage phenotypes previously reported by our lab and others of decreasing M1-associated cytokine markers with simultaneously elevated *Il10* (*Kelly et al., 2007*; *Sene et al., 2013*).

## Macrophages from mice with conditional *Mir146b* deletion demonstrated altered cell polarization

To assess the in vivo effect of the loss of macrophage *Mir146b* expression, we generated a novel *Mir146b* knockout mouse strain (*Mir146b*^flox/flox^) (*Figure 2—figure supplement 1*) and crossed it with the *Lyz2*^Cre^ mouse line in order to generate mice with conditional *Mir146b* deletion in macrophages (hereafter referred to as cKO). *Mir146b*^flox/flox^ littermates (hereafter referred to as Control) were used as controls. *Mir146b* loss was confirmed by qPCR using independent biological replicates from 3-month-old females (*Figure 2D*), and we determined that there was no compensation in expression of the *Mir146* family member *Mir146a* (*Figure 2E*), whose mature sequence differs by only two nucleotides, both located outside of the seed region.

TGEMs harvested from young (6–12 weeks old) female cKO mice displayed altered polarization on gene expression analysis (*Figure 2F*) similar to that observed with *Mir146b* knockdown in TGEMs (*Figure 2C*). Compared to TGEMs isolated from littermate Controls, cKO TGEMs expressed lower levels of *Nos2*, *Il1b*, *Il6*, and *Ccl2*, traditionally associated with classical macrophage polarization, as well as *Arg1* and *Cd163*. They also displayed a trend toward increased expression of the cytokine associated with alternative activation, *Il10*, all similar to the pattern observed following in vitro knockdown of *Mir146b*. However, *Mmp9*, which decreased following knockdown, showed an increasing trend in cKO TGEMs compared to littermate Controls. One possible explanation for this disparity may be that the life-long downregulation of *Mir146b* expression experienced by TGEMs in vivo results in different cellular stresses and additional regulation than a short-term (72 hr) partial knockdown in culture. Taken together, however, even without added activating stimulation (aside from potential effects of thioglycollate elicitation) such as IFNγ + LPS or IL4, these results suggest that deficiencies in *Mir146b* lead to abnormal cytokine gene expression. Similar patterns have previously been reported with tumor-associated macrophages and macrophages in models of age-related macular degeneration (AMD), a blinding eye disease (*Wang et al., 2019*; *Kelly et al., 2007*; *Sene and Apte, 2014*; *Sene et al., 2013*; *Mantovani et al., 2017*).

## TGEMs lacking *Mir146b* have phenotypically abnormal and functionally deficient mitochondria

In order to characterize *Mir146b*-deficient TGEMs on a subcellular level, we examined electron micrographs of cells isolated from cKO female mice and littermate Controls. Analysis revealed a decreased number of mitochondria in TGEMs lacking *Mir146b* compared to littermate Controls (*Figure 3A–C*). In addition, the mitochondria from cKO macrophages had increased intercrystal

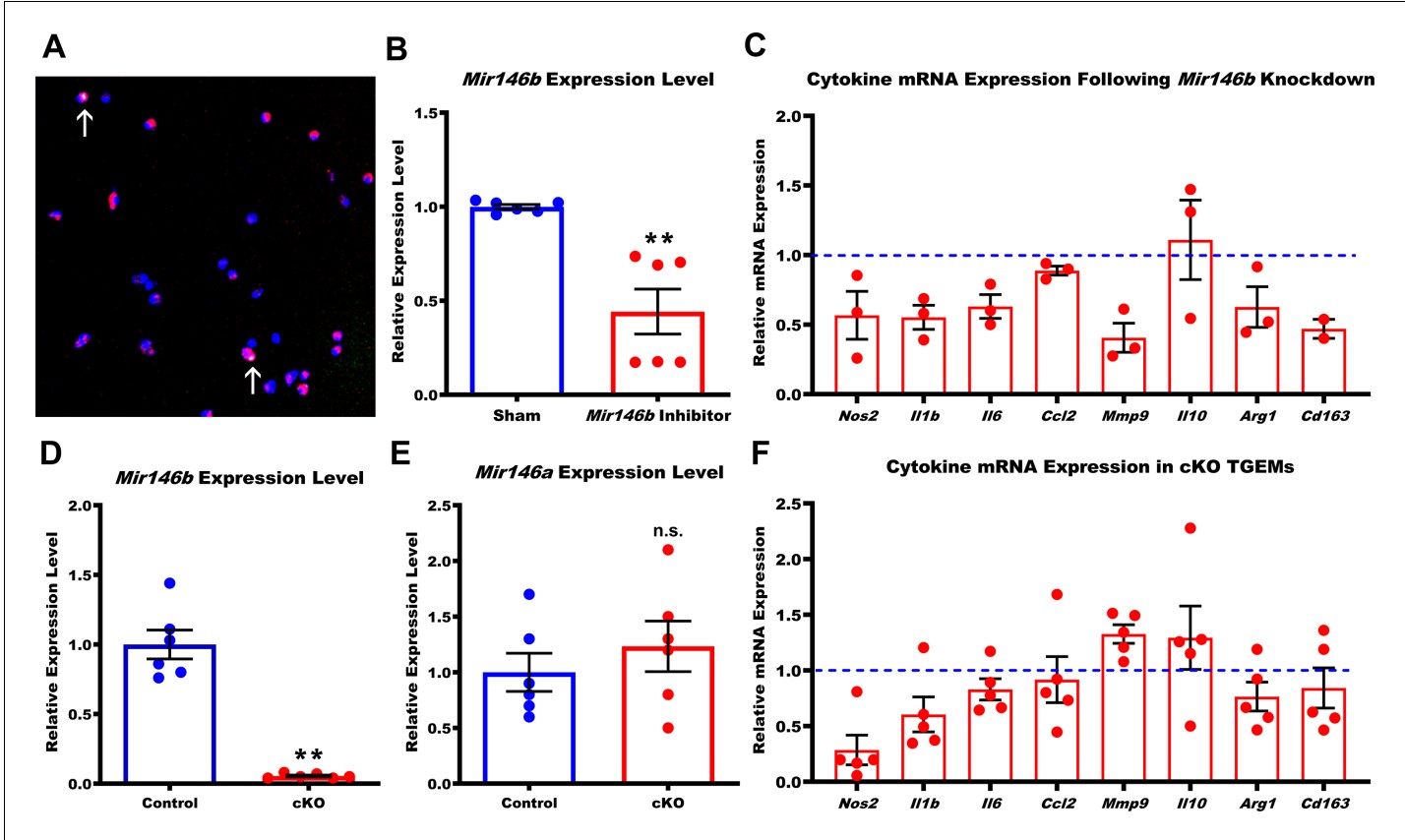

**Figure 2.** Loss of *Mir146b* in TGEMs alters inflammatory cytokine profile. (A) Fast-forward transfection of thioglycollate-elicited macrophages (TGEMs) from 3-month-old C57Bl/6 female mice with small (~22 nucleotides) RNAs (magenta) for microRNA knockdown causes little apoptosis as determined by terminal deoxynucleotidyl transferase dUTP nick end labeling (TUNEL) staining (green/arrows) while (B) producing efficient knockdown of *Mir146b* vs scramble transfection control as determined by quantitative polymerase chain reaction (qPCR) expression analysis (n = 6 vs n = 6 from two separate experiments of n = 3 independent wells each; **p<0.0022). (C) Cytokine gene expression assessed by qPCR following *Mir146b* knockdown relative to sham control (blue dash line) in TGEMs. To obtain sufficient cell numbers for each experiment, TGEMs from n = 2–3 3-month-old female mice were pooled into a single sample. Each sample was run in two to three technical replicates, and the average Ct values were used for analysis. Each dot represents the normalized expression for one independently pooled sample relative to pooled littermate Control samples. (D) TGEMs from young female conditional knockout (cKO) mice have significantly reduced *Mir146b* gene expression levels (n = 6 vs n = 6 mice [biological replicates]; **p = 0.0049) (E) but no change in *Mir146a* compared to TGEMs from female littermate Controls (n = 6 vs n = 6 mice [biological replicates]; p = 0.5738 [not significant]). (F) TGEM cytokine gene expression from young female cKO mice vs littermate Controls (represented by the blue dashed line). Each dot represents the relative gene expression in cKO TGEMs pooled from two to three female mice per sample (as in C above) compared to that of pooled TGEMs from female littermate Controls for each independent experiment (n = 5 independent experiments). Graphical data are represented as mean ± SEM. Mann-Whitney U-test was used to compare between groups for (B), (D), and (E).

The online version of this article includes the following source data and figure supplement(s) for figure 2:

**Source data 1.** Relative *Mir146b* miRNA expression values used for graphs from qPCR of sham-transfected (n = 6) or *Mir146b* inhibitor-transfected (n = 6) female TGEMs.

**Source data 2.** Relative mRNA expression values of *Nos2*, *Il1b*, *Il6*, *Ccl2*, *Mmp9*, *Il10*, *Arg1*, and *Cd163* used for graphs from qPCR of sham-transfected (n = 3 independent biological samples) vs *Mir146b* inhibitor-transfected (n = 3 independent biological samples) pooled female TGEMs.

**Source data 3.** Relative *Mir146b* miRNA expression values used for graphs from qPCR of Control (n = 6 mice) vs cKO (n = 6 mice) TGEMs.

**Source data 4.** Relative *Mir146a* miRNA expression values used for graphs from qPCR of TGEMs from female Control (n = 6) vs cKO (n = 6) mice.

**Source data 5.** Relative mRNA expression values of *Nos2*, *Il1b*, *Il6*, *Ccl2*, *Mmp9*, *Il10*, *Arg1*, and *Cd163* used for graphs from qPCR of pooled TGEMs from female Control (n = 8 independent biological samples) vs cKO (n = 7 independent biological samples) mice.

**Figure supplement 1.** Generation of mice with conditional deletion of *Mir146b* in macrophages.

spaces (ICS) compared to Control TGEMs (*Figure 3D–F*), which may affect mitochondrial respiration and metabolic function.

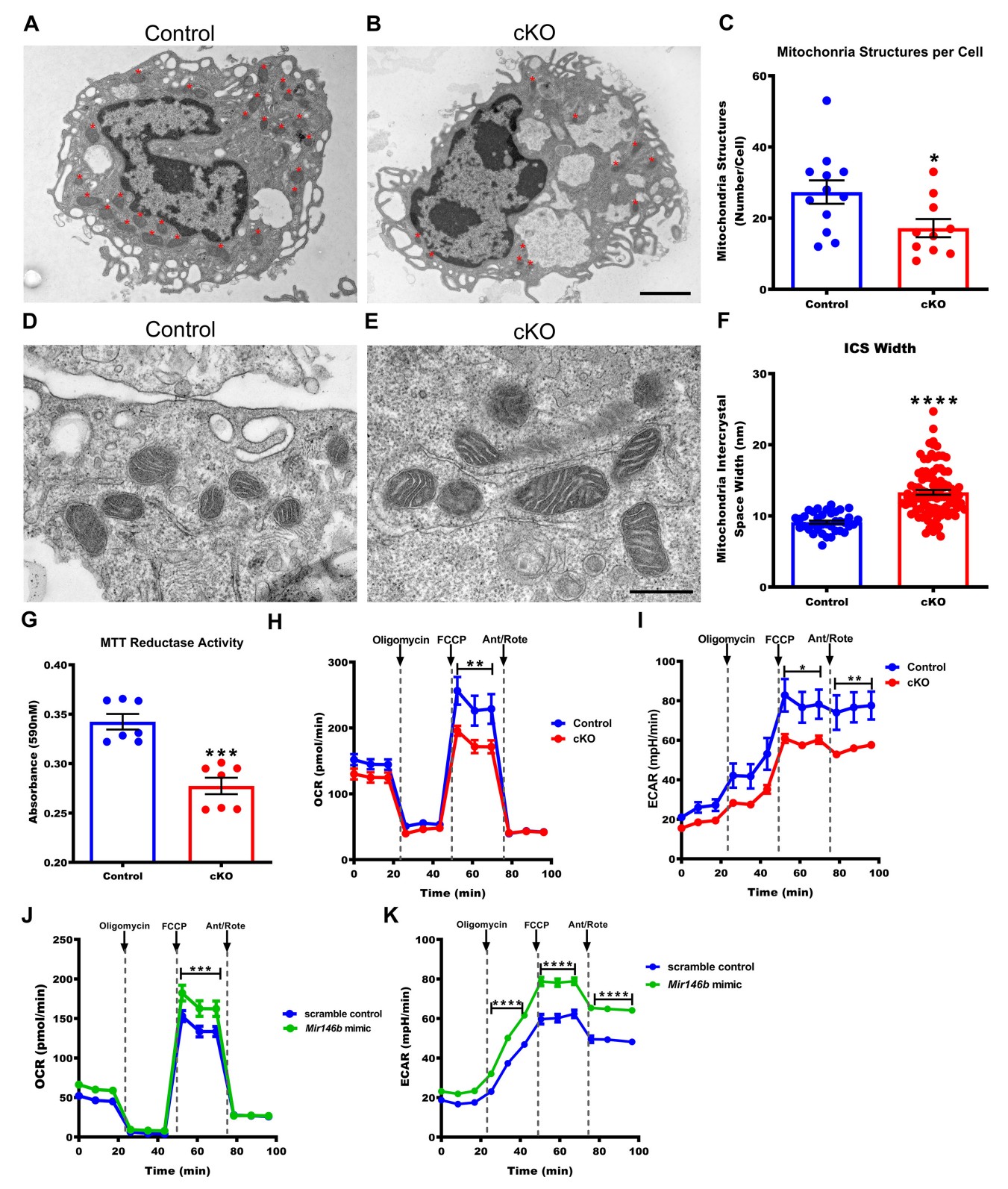

**Figure 3.** Macrophages deficient in *Mir146b* have abnormal mitochondria with decreased functional capacity. (**A**) Representative electron micrographs from Control and (**B**) conditional knockout (cKO) thioglycollate-elicited macrophages (TGEMs) from young female mice. Red asterisks indicate mitochondrial organelles. Scale bar = 2 μm. (**C**) Quantification of the average mitochondrial number per TGEM (n = 12 vs n = 10; *p = 0.0318). (**D**) Representative high-magnification electron micrograph images of mitochondria from young female Control and (**E**) cKO TGEMs. Scale bar = 500 nm.

*Figure 3 continued on next page*

Figure 3 continued

(F) Quantification of the average macrophage mitochondria intermembrane space width from young female cKO vs Control (n = 37 vs n = 100; ****p<0.0001) TGEMs. (G) MTT (3-[4,5-dimethylthiazol-2-yl]−2,5 diphenyl tetrazolium bromide) reduction by TGEMs from young (6–12 weeks old) female cKO vs Control (n = 7 vs n = 7 mice; ***p = 0.0006) TGEMs. (H) Oxygen consumption rates over time for young female cKO and Control TGEMs from representative Seahorse XF MitoStress assay (one-way analysis of variance [ANOVA], n = 3 vs n = 4 mice [biological replicates]; ****p<0.0001; Bonferroni's multiple comparison test carbonyl cyanide 4-(trifluoromethoxy) phenylhydrazone (FCCP), **p<0.01). (I) Extracellular acidification rates over time for young female cKO and Control TGEMs (one-way ANOVA, ****p<0.0001; n = 3 vs n = 4 mice [biological replicates]; Bonferroni's multiple comparison test FCCP, *p<0.05; antimycin A/rotenone (Ant/Rote), **p<0.01). (J) Oxygen consumption rates over time for young female C57Bl/6J TGEMs overexpressing *Mir146b* vs scramble control (one-way ANOVA, ****p<0.0001; n = 6 vs n = 8 biological replicates; Bonferroni's multiple comparison test FCCP, ***p<0.001). (K) Extracellular acidification rates over time for young female C57Bl/6J TGEMs overexpressing *Mir146b* vs scramble control (one-way ANOVA, ****p<0.0001; n = 6 vs n = 8 biological replicates; Bonferroni's multiple comparison test oligomycin, ****p<0.0001; FCCP, ****p<0.0001; Ant/Rote, ****p<0.0001). Data represent mean ± SEM. For (C), (F), and (G), Mann-Whitney U-test was used for comparison between groups.

The online version of this article includes the following source data for figure 3:

**Source data 1.** Representative x5000 transmission electron microscopy (TEM) images of Control (WT) TGEMs.

**Source data 2.** Representative x5000 TEM images of cKO TGEMs.

**Source data 3.** Numerical values used in graphs from the number of mitochondrial structures per cell in Control (Cntl) (n = 12) vs cKO (n = 9) female TGEMs.

**Source data 4.** Representative x25,000 TEM images of Control (WT) TGEMs.

**Source data 5.** Representative x25,000 TEM images of cKO TGEMs.

**Source data 6.** Numerical values used for graphs from measurements of intracrystal space width (nm) in TGEMs from female Control (WT) vs cKO TGEMs.

**Source data 7.** Numerical values of absorbance at 590 nM used for graphs from the MTT assay of female Control (Cntl) (n = 7 mice) vs cKO (n = 7 mice) TGEMs.

**Source data 8.** Numerical values used for graphs of the oxygen consumption rate (pmol/min) from Seahorse Mitostress Assay of TGEMs from female Control (n = 3 mice) vs cKO (n = 4 mice) TGEMs.

**Source data 9.** Numerical values used for graphs of the extracellular acidification rate (mpH/min) from Seahorse Mitostress Assay of TGEMs from female Control (n = 3 mice) vs cKO (n = 4 mice) TGEMs.

**Source data 10.** Numerical values used for graph of the oxygen consumption rate (pmol/min) from Seahorse Mitostress Assay of TGEMs from female Control transfected (n = 6) vs *Mir146b* overexpression (n = 8).

**Source data 11.** Numerical values used for graphs of the extracellular acidification rate (mpH/min) from Seahorse Mitostress Assay of TGEMs from Control transfected (n = 6) vs *Mir146b* overexpression (n = 8).

## Loss of TGEM *Mir146b* affects mitochondrial respiration

Given the abnormal number and structure of mitochondria in cKO TGEMs, we next examined the effects of loss of *Mir146b* on mitochondrial function. The tetrazolium salt 3-[4,5-dimethylthiazol-2-yl]−2,5 diphenyl tetrazolium bromide (MTT) is reduced into formazan crystals by oxioreductases, primarily (though not exclusively) in the mitochondria and provides an estimation of cell metabolism (*Berridge and Tan, 1993*). Compared to Control macrophages under standard tissue culture conditions, TGEMs from young female cKO mice reduce less MTT, indicating decreased metabolic activity (*Figure 3G*). To more specifically examine mitochondrial metabolism, we next utilized the Seahorse XF Mito Stress test to assess the oxygen consumption rate (OCR) as a measure of oxidative respiration. cKO TGEMs displayed a significant decline in oxidative phosphorylation (OXPHOS) maximal respiration, as assessed after treatment with the uncoupling agent carbonyl cyanide 4-(trifluoromethoxy) phenylhydrazone (FCCP) (*Figure 3H*), as well as decreased extracellular acidification rates (ECARs) (*Figure 3I*). Together, these results indicate that macrophages lacking *Mir146b* have decreased metabolic activity. Interestingly, when we overexpressed *Mir146b* mimic for 48 hr by transient transfection in TGEMs from young C57Bl/6J wildtype mice, the maximal mitochondrial respiration (OCR) rates and ECAR were increased compared to scramble-transfected controls (*Figure 3J–K*), providing further evidence that *Mir146b* levels influence gene expression and mitochondrial function in TGEMs.

## Loss of macrophage *Mir146b* affects gene expression

We next sought to elucidate the molecular mechanism by which *Mir146b* contributes to aging, polarization, and mitochondrial dysfunction by performing RNA-seq of TGEMs from female cKO and littermate Control (Cntl) mice using four biological replicates from each group. Although genes

upregulated in *Mir146b*-deficient macrophages are of interest as this pattern may indicate that *Mir146b* directly targets this/these gene(s) through canonical or non-canonical miRNA seed binding, we wanted to examine gene expression in a more global and unbiased manner so as to capture any significant changes in expression that may be related to mitochondrial dysfunction. We found several genes significantly downregulated in *Mir146b* cKO TGEMs, which have critical roles in both mitochondrial morphology and respiration (*Sdhd*, *Crtc2*, *Pnpt1*, *Gdf15*, *Mrps28*, *Mterf3*, *Med30 Rnaseh1*, *Slc19a2*, and *Gtpbp10*) (*Gottlieb and Tomlinson, 2005*; *Linke et al., 2017*; *Shimada et al., 2018*; *Liu et al., 2019*; *Sylvester et al., 2004*; *Taylor and Turnbull, 2007*; *Krebs et al., 2011*; *Lima et al., 2016*; *Jungtrakoon et al., 2019*; *Lavdovskaia et al., 2018*; *Figure 4A*). These findings suggest that macrophage *Mir146b* deficiency affects genes in the macrophage transcriptome that broadly regulate mitochondrial function and metabolism, leading to mitochondrial dysfunction and reduced metabolic capacity, similar to what is seen with aging (*Xia et al., 2016*; *van Beek et al., 2019*; *Pence and Yarbro, 2018*). Interestingly, only one gene, *Lyz1*, demonstrated a statistically significant increase in expression upon the loss of *Mir146b* in TGEMs. *Lyz1* is a macrophage marker that has not previously been associated with mitochondrial dysfunction, and these results could not be replicated in *mir146b* overexpression experiments, where levels of *Lyz1* were unaffected (data not shown).

## Single-cell transcriptional profiling identifies three discrete macrophage populations and reveals cell origins

TGEMs are a heterogeneous population consisting of both long-lived resident tissue macrophages as well as recruited bone marrow-derived monocytic macrophages. As our bulk RNA-seq analysis uncovered only a small number of differentially expressed genes in *Mir146b*-deficient macrophages, without a single obvious candidate to directly explain mitochondria structural and functional disparities between cKO and Control cells, we hypothesized that the heterogeneity of our samples may be masking critical differences between the two genotypes. We also asked whether increasing age may amplify these transcriptional differences. To address these questions, we performed single-cell RNA sequencing (scRNA-seq) using TGEMs isolated from young (<4 months) and old (>17 months) cKO female mice and age-matched littermate Controls. Analysis revealed three transcriptionally distinct clusters (*Figure 4B* and *Figure 4—figure supplement 1A*). All clusters expressed macrophage markers including *Mertk*, *Csf1r*, and *Cd68* (*Figure 4C*). Expression of markers including *Apoe*, *Ms4a7*, *H2-DMa*, *C3*, and *Cd74* was enriched in Cluster 1, indicating that these cells represent mature recruited monocyte-derived macrophages specialized for antigen presentation. Meanwhile, canonical biomarkers of resident peritoneal macrophages *Gata6*, *Vsig4*, *Timd4*, and *Marco* were enriched in Cluster 2 cells. Resident macrophage markers were virtually absent from Cluster 3, but these cells showed elevated expression of genes corresponding to highly active recruited macrophages including *Atp6v0d2*, *Htra1*, *Mfge8*, and *Chpt1* (*Figure 4D*).

We first sought to validate the bulk RNA-seq data and determine whether these changes were consistent across the three clusters or whether cell types were affected differently. A majority of the down-regulated genes identified by bulk RNA-seq displayed lower expression in clusters 1 and 3 of cells from cKO mice (*Figure 4E*). This loss of expression is consistent with a premature aging phenotype as many of the genes also show decreased expression in aged Control macrophages compared to young Control cells, although the resident peritoneal macrophages (Cluster 2) displayed equal or elevated expression of a subset of the genes. *Lyz1*, which was upregulated in cKO TGEMs in the bulk RNA-seq, was most highly expressed in resident peritoneal macrophages (Cluster 2). Its expression increased in cKO macrophages across all clusters compared to the Control counterparts; however, this increase was not recapitulated by natural aging (young Control vs old Control). As discussed above, we were unable to modulate *Lyz1* expression using overexpression models. Taken together, these may suggest that *Mir146b*'s role in *Lyz1* regulation is indirect. Alternatively, this discrepancy may be caused by amplifying or divergent effects due to the dramatic loss of *Mir146b* associated with our knockout model compared to the slow decline that occurs with natural aging, estimated to be approximately 25% (reduction) at this time point based upon our initial RNA-seq data (*Figure 1A*). These levels of *Mir146b* may either be sufficient to continue to regulate *Lyz1*, or the gradual rate of loss may allow for additional compensatory regulation by other factors.

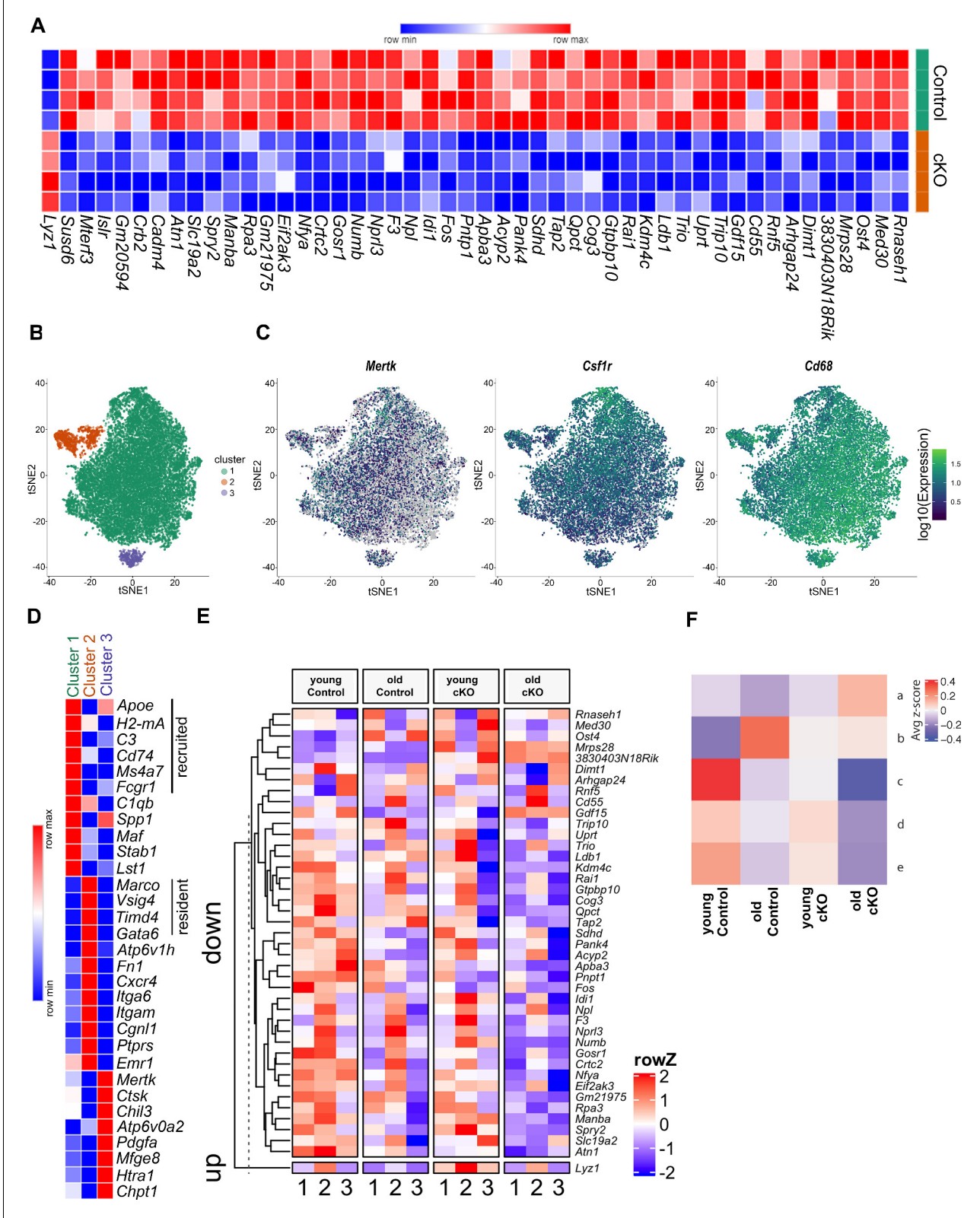

**Figure 4.** Differential gene expression in cKO TGEMs by RNA-seq and single-cell RNA-seq. (**A**) Transcriptome heatmap representation of differentially expressed genes from bulk RNA sequencing (RNA-seq) analysis of thioglycollate-elicited macrophages (TGEMs) from 3-month-old female conditional knockout (cKO) or littermate Controls (cKO n = 4 vs Control n = 4 mice [biological replicates]). (**B**) Transcription reads from single-cell RNA-seq of TGEMs from 3-month (young)- and 17-month (old)-old female cKO mice and age-matched littermate Control mice overlaid on a t-distributed Stochastic

*Figure 4 continued on next page*

*Figure 4 continued*

Neighbor Embedding (tSNE) plot separate into three distinct clusters. (**C**) Expression of macrophage markers *Mertk, Csfr1*, and *Cd68* overlaid on tSNE plots. (**D**) Heatmap comparison of resident, recruited, and activation markers across the three clusters. (**E**) Heat map distribution of reads from single-cell RNA-seq of genes found to be differentially expressed by bulk RNA-seq. (**F**) Mean row z-score across the four samples (young Control, old Control, young cKO, and old cKO) for each hierarchically defined pattern (a–e) of genes differentially expressed by cells within the main cluster, Cluster 1. Individual genes within each pattern are shown in the *Figure 4—figure supplement 2* heatmap and are listed in table form in *Supplementary file 1*. The online version of this article includes the following source data and figure supplement(s) for figure 4:

**Source data 1.** DESeq2 result file used for creating heatmaps from bulk RNA-seq of TGEMs from female Control vs cKO mice.
**Source data 2.** Normalized average expression of each expressed gene within the three clusters defined by scRNA-seq analysis of young Control, young cKO, old Control, and old cKO TGEMs from female mice.
**Source data 3.** Genes from clustering results used for creating heatmaps from scRNA-seq of young Control, young cKO, old Control, and old cKO TGEMs from female mice.
**Figure supplement 1.** Single-cell RNA-seq analysis reveals a shift with physiologic age and/or deletion of *Mir146b* in murine macrophages.
**Figure supplement 2.** Analysis of single-cell RNA-seq Cluster 1 reveals five transcriptionally distinct hierarchical subclusters.

### Differential expression analysis reveals an increase in interferon-related transcripts in *Mir146b* knockout TGEMs

Since the majority of cells belonged to Cluster 1 (*Figure 4—figure supplement 1B*), we hypothesized that transcriptional differences amongst this population of cells would likely have contributed most to the observed mitochondrial dysfunction of *Mir146b* conditional knockout TGEMs. We analyzed differential gene expression of Cluster 1 cells between the four samples (young Control, young cKO, old Control, and old cKO) and identified five (a–e) hierarchically defined expression patterns amongst our four groups (*Figure 4F*, *Figure 4—figure supplement 2*, and *Supplementary file 1*). As with our bulk RNA-seq, we again primarily noted decreases in gene expression, with four out of five patterns showing loss of gene expression in old cKO macrophages compared to age-matched controls (patterns b–e). The genes within these patterns normally display significant age-dependent changes in Control TGEMs, but the scale of expression or expression change was significantly reduced upon loss of *Mir146b*. Genes within these patterns were enriched for cellular functions such as cholesterol transport and biosynthesis (pattern b), scavenging/phagocytosis and migration (pattern c), negative regulation of inflammation (pattern d), and calcium homeostasis and endocytosis (pattern e).

Pattern a, however, highlighted genes that normally show only small decreases with age but were significantly increased in aged cKO TGEMs. Interestingly, of the 28 genes differentially expressed in pattern a, six are integral to the interferon gamma (IFNy) signaling pathway, including interferon-stimulated gene 15 (*Isg15*), C-C motif chemokine ligand 5 (*Ccl5*), interferon-induced transmembrane 3 (*Ifitm3*), interferon regulatory factor 7 (*Irf7*), interleukin-18 binding protein (*Il18bp*), and secretory leukocyte protease inhibitor (*Slpi*). These findings are especially informative as IFNy treatment has been demonstrated to significantly reduce the respiratory capacity of macrophages in culture (*Wang et al., 2018*). These data further demonstrate that a loss of *Mir146b* in TGEMs may contribute to abnormal activation, polarization, mitochondrial dysfunction, and parainflammation as seen with macrophage aging.

## Discussion

Diseases of aging are a systemic and global concern. Several cancers, atherosclerotic cardiovascular disease, neurodegenerative diseases such as Alzheimer's, and blinding eye diseases such as AMD and glaucoma are examples of diseases associated with an exponential increase in prevalence with each passing decade of life (*Nomellini et al., 2009*; *Klein et al., 2004*; *Ballard et al., 2011*; *Quigley, 2011*). Macrophage-mediated inflammation has been implicated in promoting dysfunctional immunity in many of these disorders of aging; however, the molecular mechanisms by which aging leads to dysregulated macrophage function remain unknown.

Programmatic alteration of macrophage gene expression (coding transcriptome) with age precedes the downstream effects on macrophage proteome and function. miRNAs, which are major components of the non-coding transcriptome, alter expression by either transcript degradation or translational repression (*Bartel, 2009*). Here we completed a comprehensive non-biased profiling of

the TGEM non-coding transcriptome by small RNA-seq of female mice 3–30 months of age (virtual lifespan) to probe for programmatic alterations in TGEM miRNA expression, which may contribute to the age-associated macrophage dysfunction that triggers the transition from physiologic aging to pathogenic inflammation and disease. We identified *Mir146b* and *Mir22* as microRNAs whose expression progressively decrease with age in TGEMs from female mice. *Mir146b* expression was noted to be greater than 3.3-fold higher in macrophages from young (3 months old) mice compared to *Mir22*. While future studies examining the loss of *Mir22* with aging in TGEMs are certainly warranted, here we chose to pursue the effects of *Mir146b* loss, as its initial high expression in TGEMs from young mice and gradual decline with age may represent a biologically relevant pattern consistent with the slow onset of age-related pathologies. One caveat of this study is that our original sequencing data, which identified *Mir146b* and *Mir22* as miRNAs of interest in aging TGEMs, were aligned to the mm9 version of the mouse genome. While re-alignment and re-analysis of our data using the updated mm10 genome build may reveal other interesting patterns of miRNA expression or identify new miRNAs of interest in aging macrophages, this is technically challenging due to the legacy format of the raw data, and as such, we have not been able to integrate the data into the newer build. Regardless, our subsequent qPCR data confirm that *Mir146b* expression significantly decreases from 3 to 30 months in not only TGEMs but also BMDMs. We did note that *Mir146b* is expressed at much higher levels in TGEMs, owing, perhaps, to the naive state of cultured BMDMs compared to elicited macrophages. Loss of *Mir146b* in TGEMs, established by both the transient knockdown ex vivo and a novel conditional macrophage/monocyte knockout mouse, resulted in altered cytokine expression and polarization that parallels low-grade chronic inflammation associated with aging (*van Beek et al., 2019*).

We demonstrated that *Mir146b* deletion in TGEMs results in an abnormal mitochondrial structure and dysfunctional mitochondrial metabolism, characterized by a decreased OCR and ECAR. Interestingly, this phenotype was reversed with *Mir146b* overexpression. Further examination of the effects of *Mir146b* overexpression in TGEMs as well as other murine macrophages such as BMDMs is warranted in future studies.

Perhaps, not surprisingly, given our observed TEM and Seahorse findings, we identified altered gene expression of multiple mitochondria-related genes as well as genes critical to glucose metabolism in TGEMs deficient in *Mir146b* utilizing RNA-seq. Age-related reductions in oxidative phosphorylation have been noted not only in macrophages (*Lin et al., 2018*) but also in myriad cell types, including cardiomyocytes (*Lesnefsky et al., 2016*; *Tepp et al., 2017*), intestinal mucosal epithelial cells (*Özsoy et al., 2020*), and retinal pigment epithelium of AMD patients compared to age-matched healthy controls (*Ferrington et al., 2017*), and may serve as an important indicator of age-related disease and progression. Interestingly, the only gene noted to significantly increase with *Mir146b* conditional knockout in TGEMs by bulk RNA-seq was *Lyz1*, which encodes for the bacteriolytic enzyme lysozyme C-1. This gene has not been previously linked to mitochondrial function and was not significantly increased with age in scRNA-seq or altered in overexpression experiments. Our study identifies a novel role for *Mir146b* in mitochondrial metabolic function in macrophages within the innate immune system and suggests that a gradual loss of *Mir146b* expression with age contributes to TGEM dysfunction, possibly due to an altered IFN gene response. These changes in TGEM metabolism, activation, and polarization may contribute to the para-inflammation associated with aging and diseases of aging. Further research will be necessary to determine whether this age-associated dysfunction extends to macrophages from other tissues and whether it is therapeutically modifiable.

## Materials and methods

### Key resources table

| Reagent type (species) or resource | Designation | Source or reference | Identifiers | Additional information |
|---|---|---|---|---|
| Gene (*Mus musculus*) | *Mir146b* | Ensembl | ENSMUSG00000070127 | |

*Continued on next page*

*Continued*

| Reagent type (species) or resource | Designation | Source or reference | Identifiers | Additional information |
|---|---|---|---|---|
| Strain, strain background (*M. musculus*) | *Mir146b*<sup>flox/flox</sup> | This paper (see 'Materials and methods' and *Figure 2—figure supplement 1*) | ENSMUSG00000070127 | Mouse line |
| Strain, strain background (*M. musculus*, female) | C57Bl/6 | NIA Aged Rodent Colony | C57BL/6 RRID:SCR_007317 | |
| Strain, strain background (*M. musculus*, female) | C57Bl/6 | The Jackson Laboratory | 000664 RRID:IMSR_JAX:000664 | |
| Strain, strain background (*M. musculus*, female) | *Lyz2*<sup>Cre</sup> | The Jackson Laboratory | 004781 RRID:IMSR_JAX:004781 | Mouse line |
| Cell line (*M. musculus*) | Primary thioglycollate-elicited macrophages | This paper (see 'Materials and methods') | C57Bl/6, 3–30 months; Control or cKO, 3–17 months | Freshly isolated |
| Cell line (*M. musculus*) | Primary Bone Marrow-Derived Macrophages | This paper (see 'Materials and methods') | C57Bl/6, 3–30 months; Control or cKO, 3 months | Freshly isolated |
| Transfected construct (*M. musculus*) | mirVana *Mir146b* inhibitor | Life Technologies | 4464084 | |
| Transfected construct (*M. musculus*) | mirVana negative control | Life Technologies | 4464076 | |
| Transfected construct (*M. musculus*) | miRCURY LNA *Mir146b* mimic | Qiagen | Cat # 3391173 GeneGlobe ID: YM00472354-ABD | 5'-UGAGAACUGAA UUCCAUAGGCU-3' |
| Transfected construct (*M. musculus*) | miRCURY LNA negative control A | Qiagen | Cat# 300611-04 | 5'-/F6-FAM/AACA CGTCTATACGC-3' |
| Sequence-based reagent | *hsa-Mir146b-5p* | Qiagen | Cat# 339306 GeneGlobe ID: YP00204553 | miRNA expression |
| Sequence-based reagent | *hsa-Mir146a-5p* | Qiagen | Cat# 339306 GeneGlobe ID: YP00204688 | miRNA expression |
| Sequence-based reagent | *U6* snRNA | Qiagen | Cat# 339306 GeneGlobe ID: YP00203907 | miRNA expression |
| Sequence-based reagent | *hsa-Mir-15a-5p* | Qiagen | Cat# 339306 GeneGlobe ID: YP00204066 | miRNA expression |
| Sequence-based reagent | *hsa-Mir-18a-5p* | Qiagen | Cat# 339306 GeneGlobe ID: YP00204207 | miRNA expression |
| Sequence-based reagent | *hsa-Mir-22-5p* | Qiagen | Cat# 339306 GeneGlobe ID: YP00204255 | miRNA expression |
| Sequence-based reagent | *hsa-Mir-29a-5p* | Qiagen | Cat# 339306 GeneGlobe ID: YP00204430 | miRNA expression |
| Sequence-based reagent | *mmu-Mir-362-5p* | Qiagen | Cat# 339306 GeneGlobe ID: YP00205073 | miRNA expression |
| Sequence-based reagent | *hsa-Mir-423-5p* | Qiagen | Cat# 339306 GeneGlobe ID: YP00205624 | miRNA expression |
| Sequence-based reagent | QuantiGene *Mir146b-5p* | Affymetrix | Cat# SM-10013-01 | miRNA expression |
| Sequence-based reagent | QuantiGene *U6* | Affymetrix | Cat# SR-19005-01 | miRNA expression |

*Continued on next page*

*Continued*

| Reagent type (species) or resource | Designation | Source or reference | Identifiers | Additional information |
|---|---|---|---|---|
| Sequence-based reagent | Universal *Neo* F | This paper (see *Supplementary file 2)* | Genotyping forward primer | TGC TCC TCG CGA GAA AGT ATC CAT CAT GGC |
| Sequence-based reagent | Universal *Neo* R | This paper (see *Supplementary file 2)* | Genotyping reverse primer | CGC CAA GCT CTT CAG CAA TAT CAC GGG TAG |
| Sequence-based reagent | *Mir146b Neo* F | This paper (see *Supplementary file 2)* | Genotyping forward primer | ATA TCT GGC CCA CCA GGA ACA CAT |
| Sequence-based reagent | *Mir146b Neo* R | This paper (see *Supplementary file 2)* | Genotyping reverse primer | AGC CTC TGT GTG TGC TTG TGA CAT |
| Sequence-based reagent | *LoxP* F | This paper (see *Supplementary file 2)* | Genotyping forward primer | TAA CGG CAT TAG CCA CCA CCT TCA |
| Sequence-based reagent | *LoxP* R | This paper (see *Supplementary file 2)* | Genotyping reverse primer | TGG GTT ATG TAG GGA TCC TGG GTT |
| Sequence-based reagent | *Flp/o* FWD | This paper (see *Supplementary file 2)* | Genotyping forward primer | ATA GCA GCT TTG CTC CTT CG |
| Sequence-based reagent | *Flp/o* REV | This paper (see *Supplementary file 2)* | Genotyping reverse primer | TGG CTC ATC ACC TTC CTC TT |
| Sequence-based reagent | *Flp/o* Internal FWD | This paper (see *Supplementary file 2)* | Genotyping forward primer | CTA GGC CAC AGA ATT GAA AGA TCT |
| Sequence-based reagent | *Flp/o* Internal Rev | This paper (see *Supplementary file 2)* | Genotyping reverse primer | GTA GGT GGA AAT TCT AGC ATC ATC C |
| Sequence-based reagent | *Lyz2*$^{Cre}$ F | This paper (see *Supplementary file 2)* | Genotyping forward primer | GTA GGT GGA AAT TCT AGC ATC ATC C |
| Sequence-based reagent | *Lyz2*$^{Cre}$ R | This paper (see *Supplementary file 2)* | Genotyping reverse primer | TGG GTT ATG TAG GGA TCC TGG GTT |
| Sequence-based reagent | *Actinb* gene expression assay | Life Technologies | Cat# 4352933E Assay ID: Mm00607939_s1 | mRNA expression |
| Sequence-based reagent | *Gapdh* gene expression assay | Life Technologies | Cat# 4331182 Assay ID: Mm99999915_g1 | mRNA expression |
| Sequence-based reagent | *Nos2* gene expression assay | Life Technologies | Cat# 4331182 Assay ID: Mm00440502_m1 | mRNA expression |
| Sequence-based reagent | *Mmp9* gene expression assay | Life Technologies | Cat# 4331182 Assay ID: Mm00442991_m1 | mRNA expression |
| Sequence-based reagent | *Il6* gene expression assay | Life Technologies | Cat# 4331182 Assay ID: Mm00446191_m1 | mRNA expression |
| Sequence-based reagent | *Il1b* gene expression assay | Life Technologies | Cat# 4331182 Assay ID: Mm01336189_m1 | mRNA expression |
| Sequence-based reagent | *Ccl2* gene expression assay | Life Technologies | Cat# 4331182 Assay ID: Mm00478593_m1 | mRNA expression |
| Sequence-based reagent | *Cd163* gene expression assay | Life Technologies | Cat# 4331182 Assay ID: Mm00474091_m1 | mRNA expression |
| Sequence-based reagent | *Arg1* gene expression assay | Life Technologies | Cat# 4331182 Assay ID: Mm00475988_m1 | mRNA expression |

*Continued on next page*

*Continued*

| Reagent type (species) or resource | Designation | Source or reference | Identifiers | Additional information |
|---|---|---|---|---|
| Sequence-based reagent | *Il10* gene expression assay | Life Technologies | Cat# 4331182 Assay ID: Mm00439614_m1 | mRNA expression |
| Chemical compound, drug | 3-[4,5-dimethylthiazol-2-yl]-2,5 diphenyl tetrazolium bromide | Millipore-Sigma | Cat# M5655 | MTT |
| Chemical compound, drug | Thioglycollate | Millipore-Sigma | Cat# T9032 | |
| Antibody | CD11b (Rat Monoclonal; (M1/70), eFluor 450) | ThermoFisher | 48-0112-82 RRID:AB_1582236 | Flow (1:100) |
| Antibody | F4/80 (Rat Monoclonal; (BM8), APC) | ThermoFisher | 17-4801-82 RRID:AB_2784648 | Flow (1:100) |
| Commercial assay or kit | RNeasy Mini Kit | Qiagen | Cat# 74104 | RNA extraction |
| Commercial assay or kit | mirVANA RNA Isolation kit | Life Technologies | Cat# AM1560 | RNA extraction |
| Commercial assay or kit | QuantiGene 2.0 miRNA Assay | Affymetrix | Cat# QS0008 | |
| Commercial assay or kit | miRCURY LNA RT kit | Qiagen | Cat# 339340 | |
| Commercial assay or kit | High Capacity cDNA Reverse Transcription Kit | Life Technologies | Cat# 4368813 | |
| Commercial assay or kit | ApopTag Fluorescein In Situ TUNEL labeling Kit | Millipore-Sigma | Cat# S7110 | |
| Commercial assay or kit | Seahorse XF Cell Mito Stress test kit | Agilent Technologies | Cat# 103708-100 | |
| Software, algorithm | ATM Image Capture Engine V602 software | Advanced Microscopy Techniques | | |
| Software, algorithm | Cofactor EXP software package | CoFactor Genomics | | |
| Software, algorithm | STAR aligner | GitHub | RRID:SCR_004463 | |
| Software, algorithm | Quant3p | GitHub | RRID:SCR_021236 | |
| Software, algorithm | Phantasus | Bioconductor | RRID:SCR_006442 | |
| Software, algorithm | Cellranger | 10x Genomics | RRID:SCR_017344 | |
| Software, algorithm | Monocle3 | GitHub | RRID:SCR_018685 | |
| Software, algorithm | Graphpad Prism | Graphpad | RRID:SCR_002789 | |
| Other | TopCount NTX counter | Perkin Elmer | | |
| Other | Spark multi-mode plate reader | Tecan | | |

*Continued on next page*

*Continued*

| Reagent type (species) or resource | Designation | Source or reference | Identifiers | Additional information |
|---|---|---|---|---|
| Other | Seahorse XF96 Extracellular Flux Analyzer | Agilent Technologies | RRID:SCR_013575 | |
| Other | StepOne Plus Real-Time PCR System | Life Technologies | RRID:SCR_015805 | |
| Other | Viia 7 Real-Time PCR System | Life Technologies | RRID:SCR_019582 | |
| Other | Illumina GAII sequencing platform | CoFactor Genomics | | |
| Other | Illumina HiSeq2500 sequencing platform | Centre for Applied Genomics; SickKids | RRID:SCR_001840 | |
| Other | NovaSeq S4 sequencing platform | GTAC; Washington University | RRID:SCR_001030 | |
| Other | BD FACSCanto Flow Cytometry System | BD Biosciences | RRID:SCR_018055 | |

## Animals

All animal use and experiments were approved by the Institutional Animal Care and Use Committee (IACUC) of Washington University in Saint Louis and performed according to the Washington University Animal Care and Use Guidelines. Data presented within this manuscript were obtained using female mice. C57Bl/6 mice, ranging in age from 3 to 30 months, were obtained from the National Institute on Aging (Bethesda, MA) Aged Rodent Colony. To create mice lacking *Mir146b*, a targeting vector that utilized a modified pBluescript backbone was constructed using recombineering methods (*Lee et al., 2001*). The first step was the retrieval of the entire length of the construct from the RP24-161H3 Bac vector. In the next step, the lone LoxP site, 326 bp downstream of *Mir146b*, was inserted. The last step was insertion of the LoxP and Frt flanked Neo cassette, 676 bp upstream of*Mir146b*. The construct contains a 5′ homology arm, a *Pgk*-driven Neo cassette flanked by Frt sites with a LoxP site upstream of the Neo cassette, a conditional arm with a lone LoxP site downstream of the conditional arm, and finally a 3′ homology arm. The 5′ arm starts at 3877 bp upstream of *Mir146b* and is 3201 bp in length. The conditional arm is 1118 bp and contains *Mir146b*. The 3′ arm starts 327 bp downstream of *Mir146b* and is 3117 bp in length. The linearized construct was transfected into SCC10 (129x1Sv/J) embryonic stem (ES) cells and clones were screened for G418 resistance and homologous recombination via long range PCR and Southern hybridization. Positive karyotypically normal ES clones were subsequently injected into mouse blastocysts. The resulting chimeric male mouse served as the colony founder and was bred to C57Bl/6J female mice (Jackson Laboratory, Bar Harbor; ME Stock No. 000664). Mice positive for the *Mir146b* insert were bred to B6.Cg-Tg(Pgk1-flpo)10Sykr/J hemizygous Flp deleter strain (Jackson Laboratory Stock No. 011065) to remove the Universal Neo cassette. Offspring positive for the *Mir146b* knock-in construct and negative for Frt/Universal Neo were further crossed to C57Bl/6J for subsequent generations to establish the line at >99% in the C57Bl/6J background. Genetic background analysis was performed by IDEXX BioAnalytics (Columbia, MO) using C57Bl/6J as the reference background. To create mice in which *Mir146b* was conditionally knocked out in macrophages, we crossed this line with *Lyz2*^Cre mice (Jackson Laboratory stock No. 04781) to produce *Lyz*^Cre *Mir146b*^flox/flox (cKO) and *MiR146b*^flox/flox littermate controls (Controls). The final breeding scheme of *Lyz*^Cre *Mir146b*^flox/flox X *MiR146b*^flox/flox resulted in viable fertile litters with approximately half of the pups being *Mir146b* macrophage cKOs and half, Cre-negative (non-excised) *Mir146b*-construct-positive Controls. Genotyping primers and parameters can be found in *Supplementary file 2*.

## Macrophage isolation

Adult mice were injected interperitoneally with 1.5–2 ml of sterile 4% thioglycollate (Sigma; Saint Louis, MO) as previously described (*Kelly et al., 2007*; *Nakamura et al., 2015*; *Sene et al., 2013*;

*Khan and Apte, 2008*; *Santeford et al., 2016*). At day 4 post-injection, mice were euthanized by CO$_2$ asphyxiation and macrophages were collected by peritoneal lavage in 10 ml DPBS (Gibco [ThermoFisher Scientific]; Waltham, MA). Cells were pelleted at 1000 x*g* for 10 min, DPBS was decanted, and cells were resuspended and plated in Dulbecco's modified Eagle medium (DMEM) (Gibco) containing 10% FBS (Gibco), 100 U/ml penicillin/streptomycin antibiotic cocktail (Gibco), and 2 mM L-glutamine (Gibco). Following overnight cell adherence, plates were washed two to three times with DPBS to remove non-adherent cells and complete DMEM medium was replaced for approximately 24 hr, at which time they were directly assayed or harvested for further analysis. Cells were maintained in an incubator at 37°C with 5% CO$_2$.

For BMDMs, mice were euthanized via CO$_2$ asphyxiation, and the femurs and tibia were harvested. Each bone was flushed with 5 ml DMEM using a 25 g needle and syringe to collect the bone marrow. The cell suspension was passed through a 100 µM strainer to remove clumps. Cells were plated in a differentiation medium consisting of DMEM with 10% FBS, 1% L-glutamine, 1% pen/strep, 1% sodium pyruvate, and 20% conditioned medium from L929 cell culture. Cultures were washed in DPBS every 3 days and the differentiation medium replaced through d7, after which time they were switched to a medium without the addition of L929-conditioned media and prepared for assay.

## Flow cytometry

Cells were harvested by peritoneal lavage of thioglycollate-elicited female mice, aged 3 or 20 months, cultured overnight in complete DMEM to allow macrophage/monocyte cells to attach, washed with DPBS to remove non-adherent cells, and returned to culture to rest overnight as described above. To collect the adherent cells for analysis, plates were washed twice with DPBS to remove the residual culture medium and fresh ice-cold FACS buffer (1% bovine serum albumin and 0.05% sodium azide in DPBS) was added to the plate. Cells were gently removed using a cell lifter. The resulting cell suspension was pipetted up and down to achieve single cells and then passed through a 50-µM filter. Cells were stained with CD11b eFluor450 and F4/80 APC (eBiosciences) and 30,000 cells per sample were analyzed using a BD FACSCanto Flow Cytometry system.

## Small RNA-seq of C57Bl/6 macrophages

For small RNA-seq to examine miRNA expression, TGEMs were harvested, as described above, from C57Bl/6 female mice of ages 3, 6, 12, 18, 24, and 30 months (National Institute on Aging). Peritoneal exudates containing TGEMs were pooled from 10 mice per age group and plated and washed as described above. RNA (>10µg) was isolated from macrophage samples using the mirVana RNA isolation kit (ThermoFisher Scientific) as per manufacturer's instructions. RNA was randomly fragmented and converted to complementary DNA (cDNA) for sequencing using the Illumina GAII (San Diego, CA) platform. The resulting sequencing reads were analyzed by Cofactor Genomics (St Louis, MO), in consultation with the Washington University Genome Center, using the Cofactor Genomics EXP software package. Sequences were first aligned against the mouse genome (July 2007 [NCBI37/mm9]). Overlapping reads were then clustered together to assemble expressed loci and provide respective read counts and coverage for each locus. All counts and expression levels were normalized down to the sample with the fewest reads in order to allow cross sample comparative expression between loci. A x6 coverage multiplier was used as a cutoff for including reads to compensate for stochastic deep sequencing. A pair-wise comparison was performed between samples and log2 ratios were computed for each expressed small RNA. Sequences aligning to the annotated regions of the mouse genome that have been previously identified as 572 individual microRNAs were used for further analysis. We next identified microRNAs whose expression either consistently increased or decreased (ie, unidirectional change) across time in the progressively aging macrophage, allowing for +/- 10% error in the expression ratios between any two consecutive time points, such that if the (n + 1) time point compared to (n) was 0.9 < (Exp(n+1))/Exp(n) < 1.1, then it was considered to be no change and not contributing to either direction. A heatmap of the top 100 expressed miRNAs was constructed using Phantasus build 1.9.2 (*Zenkova et al., 2021*).

## miRNA expression profiling

For qPCR analysis of miRNA expression, we isolated RNA from TGEMs using the miRvana RNA isolation kit and prepared cDNA using the miRCURY LNA Universal RT microRNA PCR, Polyadenylation and cDNA synthesis kit II (Qiagen) with 65 ng of starting RNA per reaction. We performed qPCR using ExiLENT SYBR Green master mix (Qiagen) and miRCURY LNA miPCR primer sets (Qiagen). To analyze the data, we used the $\Delta\Delta$Ct method, normalizing to *U6* expression. Ct values greater than 36 were excluded from analysis. Relative miRNA expression was calculated using the average values obtained from the appropriate control group for each experiment (young [3 months] mice, sham-transfected controls, or Control mice). The following primer sets were utilized: *hsa-Mir146b-5p* (cat. # 339306, GeneGlobe ID YP00204553), *hsa-Mir22-5p* (cat. # 339306, GeneGlobe ID YP00204255), *hsa-Mir15a-5p* (cat. #339306, GeneGlobe ID YP00204066), *hsa-Mir-29a-5p* (cat. #339306, GeneGlobe ID YP00204430), *hsa-Mir423-5p* (cat. #339306, GeneGlobe IDYP00205624), *hsa-Mir146a-5p* (cat. #339306, GeneGlobe ID YP00204688), *hsa-Mir18a-5p* (cat. #339306, GeneGlobe ID YP00204207), *hsa-Mir362-5p* (cat. #339306, GeneGlobe ID YP0025073), and *U6* snRNA (hsa, mmu) (cat. #339306, GeneGlobe ID YP00203907).

## *Mir146b* expression analysis by QuantiGene Assay

TGEMs were isolated and cultured as described above. RNA was extracted using mirVana RNA isolation kit. QuantiGene 2.0 miRNA Assay (Affymetrix; Santa Clara, CA) was performed according to the manufacturer's instructions using 250 ng of RNA per reaction using *Mir146b-5p* probe set, positive control (#SM-10013–01), and *U6* (#SR-19005–01). Briefly, diluted probe sets, samples, and controls were added to wells of the 96-well capture plate, sealed, and incubated overnight at 46°C. The following morning, plates were washed 3x with Wash Buffer, and Pre-Amp solution was applied to all wells for 60 min at 46°C. After washing, 2.0 Amplifier solution was applied to each well for 60 min at 46°C, followed by additional washes and application of the Label Probe for an additional 60 min at 46°C. After a final set of washes, luminescent 2.0 Substrate was added to each replicate well and luminescence was measured on a TopCount NTX counter.

## *Mir146b* transient knockdown

TGEMs were harvested from 3-month-old C57/Bl6 female mice. For inhibition of *mmu-Mir146b*, $5 \times 10^5$ cells in 600 ul media were plated in each well of a six-well plate and allowed to adhere. Lipofectamine RNAi MAX (ThermoFisher) was diluted 1:100 in a serum-free medium and combined 1:1 with a medium containing *mirVana Mir146b* inhibitor or negative control (Life Technologies, 4464084 or 4464076). This mixture was incubated at room temperature (RT) for approximately 15 min to allow for lipofectamine/oligo complexes to form, per manufacturer's instructions. 400 ul of each solution was then added to cells for a final concentration of 2 ul lipofectamine and 25 nM inhibitor or negative control per well in 1 ml total volume. To assess transfection efficiency, additional cells were transfected using BLOCK-iT Alexa Fluor Red Fluorescent Oligo (ThermoFisher) using the same protocol. After 24–72 hr, cells transfected with BLOCK-iT were processed for TUNEL staining as described below, and cells transfected with *Mir146b* inhibitor or control sequence were harvested and processed for RNA/miRNA isolation using mirVANA microRNA isolation kit according to the manufacturer's instructions (ThermoFisher).

## TUNEL

Chemicon ApopTag Fluorescein In Situ TUNEL labeling kit (Millipore/Sigma, Burlington, MA) was used according to the manufacturer's instructions to detect apoptotic cells in cultured primary macrophage samples. Briefly, TGEMs were washed in DPBS and fixed in 1% paraformaldehyde for 10 min at RT and post-fixed in precooled ethanol:acetic acid (2:1) for 5 min at −20°C, with phosphate-buffered saline (PBS) washes before and after this step. Next, equilibration buffer was applied at RT followed by incubation with TdT enzyme at 37°C for 1 hr, and then a 10-min wash in Stop/Wash buffer. After washing in PBS, anti-digoxigenin conjugate was applied for 30 min, followed by PBS washes, counterstaining with 4 ′, 6-diamidino-2-phenylindole (DAPI), and fluorescent imaging.

## mRNA expression

For mRNA expression analysis, TGEMs were pooled from two to three female mice for each sample. We isolated RNA by using the RNeasy Plus Mini Kit (Qiagen) as per manufacturer's instructions. We prepared cDNA using the High Capacity Reverse Transcription kit (Thermo Fisher Scientific) and performed qPCR using TaqMan Fast Advanced Master Mix (Thermo Fisher Scientific) with n = 2 technical replicates per sample. We used the $\Delta\Delta Ct$ methods and normalized to the geometric mean of *Actinb* and *Gapdh* housekeeping genes. The following TaqMan Gene Expression probes were utilized: *Actinb* (Mm00607939_s1), *Gapdh* (Mm99999915_g1), *Nos2* (Mm00440502_m1), *Mmp9* (Mm00442991_m1), *Il6* (Mm00446191_m1), *Il1b* (Mm01336189_m1), *Ccl2* (Mm00478593_m1), *Cd163* (Mm00474091_m1), *Arg1* (Mm00475988_m1), and *Il10* (Mm99999062). Each experiment was conducted three to five times using independent samples.

## Transmission electron microscopy

For ultrastructural analyses, TGEMs from female cKO or Control mice were fixed in 2% paraformaldehyde/2.5% glutaraldehyde (Polysciences Inc, Warrington, PA) in 100 mM sodium cacodylate buffer, pH 7.2, for 1 hr at RT. Samples were washed in sodium cacodylate buffer and post-fixed in 1% osmium tetroxide (Polysciences Inc) for 1 hr. Samples were then rinsed extensively in distilled water ($dH_2O$) prior to en bloc staining with 1% aqueous uranyl acetate (Ted Pella Inc, Redding, CA) for 1 hr. Following several rinses in $dH_2O$, samples were dehydrated in a graded series of ethanol and embedded in Eponate 12 resin (Ted Pella Inc). Sections of 95 nm were cut with a Leica Ultracut UCT ultramicrotome (Leica Microsystems Inc, Bannockburn, IL), stained with uranyl acetate and lead citrate, and viewed on a JEOL 1200 EX transmission electron microscope (JEOL USA Inc, Peabody, MA) equipped with an AMT 8.0 megapixel digital camera and AMT Image Capture Engine V602 software (Advanced Microscopy Techniques, Woburn, MA).

## MTT reduction assay

TGEMs from female mice were plated at 100,000 cells per well in 96-well plates in complete DMEM medium as described above. At the time of the experiment, cells were washed with DPBS and DMEM medium containing 0.5 mg/ml MTT (Millipore-Sigma) was applied to the cells for 3 hr at 37°C in a 5% $CO_2$ incubator. Freshly prepared isopropanol containing 10% triton-X 100 and 0.1 N HCl was used to dissolve formazan crystals formed during the incubation. Absorbance at 570 nM was measured on a Tecan Spark multi-mode plate reader (Morrisville, NC).

## Seahorse Mito Stress Assay

For metabolic characterization of macrophages, we used the Seahorse XF Cell Mito Stress test on an XF96 Extracellular Flux Analyzer (Agilent Technologies, Santa Clara, CA) to measure the OCR as a surrogate marker for oxidative respiration. Macrophages were plated in Seahorse XF96 cell culture microplates (Seahorse Bioscience) at 100,000 cells per well. On the morning of the experiment, we washed the cells and replaced the medium with Seahorse assay medium (Agilent Technologies) supplemented with 10% FBS, 25 mM glucose (Millipore Sigma, St Louis, MO), and 1 mM sodium pyruvate (Thermo Fisher Scientific) and the pH adjusted to 7.4. After incubation in a non-$CO_2$ incubator at 37°C for 1 hr, we measured the OCR at baseline and after sequential treatment with the following chemicals from the Mito Stress Test kit (Seahorse Bioscience): 3 μM oligomycin, 5 μM FCCP, and 1 μM rotenone/antimycin A (rot/AA). Each cycle consisted of 2 min of mixing and a 1 min pause, followed by a 5-min measurement period; we repeated each cycle three times. We normalized the background of all measurements by subtracting the average OCR of each sample after treatment with rot/AA. Values from n = 6 technical replicates were averaged for each biological replicate sample.

## MicroRNA overexpression

Transient overexpression of *Mir146b* in TGEMs was achieved by fast-forward transfection of cells harvested from C57Bl/6J (Jackson) female mice using fluorescein amidite-labeled (FAM-labeled) *hsa-146b-5p* miRCURY locked nucleic acid (LNA) miRNA Mimic (5'-UGAGAACUGAAUUCCAUAGGCU-3'; Qiagen; cat. #339173 YM00472354-ABD) or negative control A LNA (5'-/F6-FAM/AACACGTCTA TACGC-3'; Qiagen; cat. #300611–04). Briefly, after harvest, macrophages were seeded at 1 x 10⁶

cells/well in 96-well plates and allowed to adhere for 2 hr. Following DPBS washes to remove non-adherent cells, transfection complexes were prepared containing 50 nM of mimic (or negative control LNA) and 0.75% HiPerFect transfection reagent (Qiagen) and incubated for 15 min at RT before adding to the cells in DMEM supplemented with 10% FBS, 1% pen/strep, and 1% L-glutamine. FAM-labeled LNA allowed us to visualize the transfection efficiency by observing with a fluorescent microscope. Cells were assessed by Seahorse XF MitoStress Assay (as described above) 48 hr post-transfection.

## RNA-seq of *Mir146b*-deficient macrophages and controls

For bulk RNA-seq, TGEMs were harvested from 3-month-old female cKO and littermate Controls. Each mouse served as an independent biological replicate. Cells were plated and harvested as described above. RNA was isolated using RNeasy mini kit (Qiagen; Germantown, MD). mRNA was extracted with oligodT beads (Life Technologies), and cDNA and libraries were constructed as previously described (*Bambouskova et al., 2018*). Libraries were sequenced at the Centre for Applied Genomics (SickKids, Toronto) using a HiSeq2500 (Illumina) 50 x 25 bp pair-end sequencing. Fastq files for each sample were aligned to the mm10 mouse genome assembly using STAR aligner. Aligned reads were quantified using quant3p script to account for specifics of 3′ sequencing with a protein coding subset of Gencode genome annotation. Differential expression analysis was determined by DESeq2 using the top 12,000 expressed genes. Heatmaps were constructed using Phantasus build 1.9.2 (*Zenkova et al., 2021*).

## scRNA-seq

For scRNA-seq, TGEMs were harvested from female cKO and littermate Control mice at 3 (15 weeks) or 17 months of age and plated as described above. Cells were profiled using the 10XGenomic platform using the 3′V3 chemistry and libraries sequenced on the Illumina NovaSeq S4 at the Genome Technology Access Center (GTAC) at Washington University.

Data were analyzed first using Cellranger 3.1.0 and mapped to the mouse mm10 genome with default parameters. Expression matrices were reanalyzed using Monocle3 (v0.2.1) (*Cao et al., 2019*). Initial clustering identified minor contamination (<1%) of cells not expressing macrophage markers (*Emr1*, *Mertk*, *Csf1r*, and *Cd68*) that were removed from further analyses. tSNE-dimension reduction was performed on the top principal components learned from the 317 genes with the highest variance across the cells that passed quality filters (>5000 mRNA counts), and the Louvain method was used to define clusters. The Monocle3 `fit_models()` function was used to assess differential gene expression with the following parameter: `model_formula_str = '~library + 1'`.

## Data availability

All sequencing data discussed herein have been deposited in NCBI's Gene Expression Omnibus (*Edgar et al., 2002*) under record number GSE164476.

STAR aligner is available at https://github.com/alexdobin/STAR (*Dobin, 2021*) and quant3p script is available at https://github.com/ctlab/quant3p (copy archived at swh:1:rev:be9977925e9e842cc755f14ced72bbee5c5d6d77; *Sergushichev, 2021a*).

Phantsus is available at https://github.com/ctlab/phantasus (copy archived at swh:1:rev:c6bb0e960554a23eb712690cbd7f8e3f7d79ca0d; *Sergushichev, 2021b*).

Cellranger is available at https://github.com/10XGenomics/cellranger (*Marks, 2021*).

Monocle3 is available at https://github.com/cole-trapnell-lab/monocle3 (*brgew, 2021*).

## Statistical analysis

Data are presented as mean ± SEM. Statistical evaluations were performed using GraphPad Prism Software version 9.1 (GraphPad, San Diego, CA). One-way mixed analysis of variance (ANOVA) with Bonferroni post-test or non-parametric Mann-Whitney U-test was used for comparison between groups. The accepted level of significance for all tests was $p < 0.05$.

## Acknowledgements

This work was supported by NIH grant R01 EY019287-08 (RSA), P30 EY02687 (Vision Core Grant), Glenn Foundation for Medical Research Award (RSA), American Federation for Aging Research Julie Martin Mid-Career Award (RSA), Carl Marshall Reeves and Mildred Almen Reeves Foundation Award (RSA), Jeffery T Fort Innovation Fund (RSA), the Starr Foundation (RSA), and an unrestricted grant from Research to Prevent Blindness to the John F Hardesty, MD Department of Ophthalmology and Visual Sciences at Washington University School of Medicine in Saint Louis. We would like to thank Jonathan B Lin, MD, PhD, for his assistance with statistical analysis.

## Additional information

### Funding

| Funder | Grant reference number | Author |
| --- | --- | --- |
| National Institutes of Health | R01 EY019287-08 | Rajendra S Apte |
| Glenn Foundation for Medical Research | | Rajendra S Apte |
| American Federation for Aging Research | | Rajendra S Apte |
| Carl Marshall and Mildred Almen Reeves Foundation | | Rajendra S Apte |
| Jeffery T. Fort Innovation Fund | | Rajendra S Apte |
| Starr Foundation | | Rajendra S Apte |
| National Institutes of Health | P30 EY02687 | Rajendra S Apte |

The funders had no role in study design, data collection and interpretation, or the decision to submit the work for publication.

### Author contributions

Andrea Santeford, Data curation, Formal analysis, Investigation, Writing - original draft, Writing - review and editing; Aaron Y Lee, Conceptualization, Formal analysis; Abdoulaye Sene, Conceptualization, Data curation, Formal analysis, Investigation; Lynn M Hassman, Maxim N Artyomov, Formal analysis, Investigation; Alexey A Sergushichev, Formal analysis; Ekaterina Loginicheva, Data curation, Investigation; Philip A Ruzycki, Formal analysis, Visualization, Writing - original draft, Writing - review and editing; Rajendra S Apte, Conceptualization, Resources, Supervision, Funding acquisition, Validation, Methodology, Writing - original draft, Project administration, Writing - review and editing

### Author ORCIDs

Andrea Santeford https://orcid.org/0000-0002-7691-6213
Rajendra S Apte https://orcid.org/0000-0003-2281-2336

### Ethics

Animal experimentation: All animal use and experiments were approved by the Institutional Animal Care and Use Committee (IACUC) of Washington University in Saint Louis and performed according to the Washington University Animal Care and Use Guidelines (protocol numbers 2018-0160 and 20-0003).

### Decision letter and Author response

Decision letter https://doi.org/10.7554/eLife.66703.sa1
Author response https://doi.org/10.7554/eLife.66703.sa2

## Additional files

### Supplementary files

• Supplementary file 1. Genes associated with each subcluster pattern from scRNA-seq analysis, graphically represented in *Figure 4—figure supplement 2*.

• Supplementary file 2. Primers used for genotyping.

• Transparent reporting form

### Data availability

Sequencing data have been deposited in GEO under accession code GSE164476.

The following dataset was generated:

| Author(s) | Year | Dataset title | Dataset URL | Database and Identifier |
|---|---|---|---|---|
| Apte R, Santeford A, Sergushichev AA, Ruzycki PA | 2021 | Loss of macrophage miR-146b with aging contributes to inflammation and mitochondrial dysfunction | https://www.ncbi.nlm.nih.gov/geo/query/acc.cgi?acc=GSE164476 | NCBI Gene Expression Omnibus, GSE164476 |

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
