## [Decision Letter]

**Acceptance summary:**

Age-related inflammation, also known as inflammaging, is a leading driver of multiple aging-related diseases. However, the molecular mechanisms underpinning why immune cells become dysfunctional during aging remain poorly understood. This study is a timely investigation demonstrating that aging leads to differential expression of micro RNAs in elicited peritoneal macrophages, which in turn promotes altered macrophage gene expression, polarization, and mitochondrial dysfunction. A key strength of this manuscript is the discovery of a role for miR-146b in macrophage biology, and the role of its decreased expression with age in driving aspects of macrophage functional decline with aging.

**Decision letter after peer review:**

Thank you for submitting your article "Loss of macrophage miR-146b with aging contributes to inflammation and mitochondrial dysfunction" for consideration by *eLife*. Your article has been reviewed by 3 peer reviewers, one of whom is a member of our Board of Reviewing Editors, and the evaluation has been overseen by Matt Kaeberlein as the Senior Editor. The reviewers have opted to remain anonymous.

Essential revisions:

1) The authors use a type of "elicited" macrophages, Thioglycollate-elicited macrophages (TGEMs), which do not represent a naive state, but an activated/recruited state. However, this information is only included in the material and methods, and not discussed in the rest of the manuscript. Since this could have a great impact on the results, the three reviewers agreed that it is essential that authors explicitly address the use of TGEMs, (even if only textually), including in the title, abstract and main text, to make sure that the narrower scope of findings is clear to readers without needing to read the material and methods. If possible (maybe for future studies), similar analyses of other macrophage types would help understand the general relevance of the finding on the impact of miR-146b on inflamm-aging.

2) Since the authors opted for an adherence-only method of purification for the TGEMs, it is crucial that some measurement of the purity of the macrophage population be provided to make sure that the purity of TGEMs by adherence is not affected by aging. An F4/80 and Cd11b flow cytometry staining on cells purified similarly from the same ages and sexes would be the ideal method for this.

3) The authors need to carefully edit the manuscript to include all relevant and necessary methodological details (e.g. sex of used mice in general and by panel, a systematic clarification of the use of technical vs. biological replicates, etc.). The authors also should refer to the individual reviewer comments for the points needed clarification in the revised manuscript on this point.

4) Generally, the authors need to improve and amend their statistical analyses. This includes (i) providing more information on the analysis leading to only miR-146b (since referees note that other miRs look significant in the analysis), (ii) removal of all t-tests since there is no power to test for data normality, removal of statistical tests when the authors only have n = 2, etc.

5) Finally, potentially contradictory findings between figures need to be reconciled or explicitly discussed by the authors. (e.g. Reviewer #2 point 4)

*Reviewer #1:*

In this manuscript, Santeford et al. study the regulation and impact of a microRNA, miR-146b, on macrophage aging phenotypes in mice. They first identify miR-146b as the only significantly and monotonously age-regulated miRNA in thioglycolate-elicited peritoneal macrophages (TGEMs) using a small RNA-seq approach. They then proceed to perform short-term manipulation of miR-146b expression in culture, as well as using a myeloid KO in vivo, and observe consistent remodeling of gene expression of cytokine genes in TGEMs. They also observe that miR-146b deficiency impacts TGEM metabolic function, including mitochondrial respiration. Transcriptome-wide analysis (both bulk and single-cell level) of the impact of miR-146b deficiency in TGEMs from the myeloid miR-146b KO model reveals decreased expression of metabolic genes and increases in interferon-signaling genes, similar to what is reported with aging.

This article identifies an interesting new regulator [miR-146b] of macrophage phenotypes with potential relationship with aging, thus providing an interesting mechanistic insight into remodeling of macrophage function with aging. Consequently, this study is of interest to the fields of macrophage biology, immunometabolism and aging biology. While the strength of this manuscript is the discovery of a role for miR-146b inn macrophage biology, and potential link with functional decline with aging, a number of points need to be addressed before the study is presented with its full context (i.e. choice of use of TGEMs) and some technical questions are answered.

A major caveat that will need to be discussed and addressed by the authors is the use of TGEMs vs. naïve/resident peritoneal macrophages with aging. Indeed, thioglycolate elicitation recruits new non-resident macrophages to the peritoneum, and will also drive them to a more activated state in response to the foreign signal. TGEMs contain a lot more "small" peritoneal macrophages (bone-marrow derived) than steady state peritoneal macrophages (usually composed of a majority of "large" peritoneal macrophages, which are resident and of embryonic origin). Although the study of TGEMs is interesting (and provides insights into populations recruited upon infection) they do not represent the steady state. This point should be very clearly stated in the text rather than just in the material and methods, at least the first time the macrophages are mentioned rather than just refer to these as "peritoneal macrophages" which is incorrect and somewhat misleading.

1) Peritoneal Macrophages used from these experiments are from thioglycolate-elicited peritoneal macrophages (majorly bone-marrow derived and recruited upon acute irritation), which are very different from the steady state population (embryonic derived and usually resting). Since thioglycolate is a way to mimic a chronic infection, some of the described biology (including the difference between TGEMs and BMDMs) could be due to a difference in response to the chronic infection instead of the difference between microRNA expression of aged animals. Thus:

a. This caveat needs to be explicitly addressed in the text the first time the authors mention the macrophages (line 80 "peritoneal macrophages"). This term should be systematically replaced with "TGEM" for accuracy throughout the manuscript.

b. The authors should discuss the caveat that effects may differ if unstimulated resident peritoneal macrophages had been evaluated instead.

c. Another point that should be discussed would be that the difference in age-regulated expression between TGEMs and BMDMs may be due to the effect of thioglycolate induction on the studied macrophages.

2) A number of studies are done with N = 2 samples (1E, 2C, 2F). This is problematic for several reasons: (i) statistics cannot be reliably applied to such small sample numbers and are thus meaningless and (ii) the use of the student t-test is for sure unwarranted, since a goodness of fit test is impossible to perform to confirm normality of data. If these pieces of data are retained as is, I recommend commenting on fold changes and nothing else. If the authors want to discuss statistics, additional samples need to be included and non-parametric tests should be used. In generally, the authors should revise analyses to use non-parametric tests instead of the Student t-test.

3) Aging is a very sex-dimorphic process, and thus a variable of interest in any study including aging as its topic. However, the sex of used animals is not given in the manuscript. Please update the manuscript to include this information. If only one sex was used, please discuss how results may be different in the other sex in the Discussion section. If both sexes were mixed, please make sure to color-code data points to differentiate females and males on the graphs.

4) Methodological details need to be included or revised for consistency reproducibility.

a. Please include a table with the sequences of all used qPCR and genotyping primers.

b. Some analyses are performed on the mm9 mouse genome build (e.g. small RNA-seq seq, line 449) and some on the mm10 genome build (e.g. RNA-seq of KO TGEMs line 591). Since this could lead to differences in results, please harmonize analyses so they are all performed on the same genomic build.

c. Please include all code/scripts used for the analysis in a supplementary document or deposit them to a Github repository as per the journal policy.

*Reviewer #2:*

Santeford et al. investigated age-dependent molecular changes in macrophages that could contribute to inflammaging. Using RNA-seq analysis on peritoneal macrophages from mice of various ages, they identified miR-146b as a microRNA that progressively and unidirectionally declined with age. Using miR-146b antagomirs (inhibitors) and conditional knockout mice, they show that the loss of miR-146b function alters the expression of several inflammatory cytokines. Microscopic analysis revealed abnormal mitochondrial morphology in thioglycolate-induced peritoneal macrophages that lack miR-146b, which was coupled with reduced maximum respiratory capacity and glycolytic rate. Single cell RNA-seq on peritoneal macrophages revealed distinct clusters and a subset with altered interferon γ signaling.

The study demonstrates an interesting role for miR-146b as a regulator of macrophage function. It provides a strong correlation between microRNAs and inflammaging, but a causal relation is yet to be established. The study surveys the expression of macrophage miR-146 from mice of various age groups, uses a unique mouse model capable of conditionally knocking out miR-46b, and scRNA-seq to dissect the complex population of peritoneal macrophages in a peritonitis model. Some limitations of the study lie on the lack of mechanistic connection between miR-146b and mitochondrial/metabolic alternations, the incomplete understanding whether aging macrophages lose the adaptive capacity to induce miR-146b upon stimulation or the ability to maintain baseline expression, and the unclear role of resident vs recruited macrophages in inflammaging.

1. The authors state that miR-146b is the only microRNA that progressively and unidirectionally declined with age (lines 25-27, lines 66-68). This is confusing as Supplementary Figure 1 shows other miRNAs that decline with age, such as miR15a. Please clarify and/or rephrase.

2. The thioglycolate-induced peritonitis model provides activated macrophages (both resident and recruited monocyte-derived macrophages). In contrast, bone marrow-derived macrophages that were differentiated in vitro are naive. This difference may underly the inconsistent in age-dependent miR-146 expression pattern in Figure 1. Thus, it is possible that what is lost during aging in macrophages is the adaptive capacity to induce miR-146b rather baseline expression. Would miR-146b levels decline with age also in bone marrow-derived macrophages if they were to be stimulated (i.e. loss of adaptive capacity to induce miR-14b)?

3. The manuscript should clearly discuss that data on thioglycolate-induced peritoneal macrophages reflect induced responses rather than naive conditions.

4. The cytokine panel in Figure 2C (miR-146b knockdown) does not include some cytokines in 2F (miR-146b knockout) – were they measured? Also, please comment on the Mmp9 expression, which is decreased in 2C but increased in 2F.

5. It would strengthen the paper if the levels of secreted cytokines (proteins) upon loss of miR-146b were measured.

6. In figure 3, the authors overexpress miR-146b and show increased mitochondrial respiration. Does this also alter the expression of cytokines measured in Figure 2?

7. The loss of miR-146b reduces OCR/ECAR and, conversely, its overexpression increases OCR/ECAR. Further, the authors show that the loss of miR-146b affects the expression of mitochondria-associated genes. On that line, does miR-146b overexpression affect similar genes (which would revert the metabolic phenotype)?

8. The miR-146b-dependent metabolic shift may result from alterations of multiple metabolic pathways that consequently affect OCR/ECAR, such as glucose metabolism. Were there metabolic genes that changed in the RNA-seq? If so, is/were there a coherent metabolic pathway(s) that is/are highlighted? If possible, quantifying metabolites that are highly relevant to macrophage function would provide further insight.

9. Are the cytokines measured in Figure 2 reflected in the scRNA-seq of Lyz2; miR-146bM-/M- mice?

10. As the authors state, peritoneal macrophages consist of a heterogeneous population of resident and recruited (monocyte-derived) macrophages. Further, monocyte-derived macrophages may not display age-dependent loss of miR-146b (Figure 1C). The authors may want to add some discussion on the potentially differential role of resident vs recruited macrophages in inflammaging. Further, have the authors tried to compare resident vs recruited macrophages in the scRNA-seq on peritoneal macrophages in addition to the 3 clusters (it is not clear whether the largest cluster 1 is a mix of both populations)?

*Reviewer #3:*

The manuscript by Santeford et al., investigates whether micro RNAs regulate macrophage function during the aging process. The authors approached this question by using an unbiased non-coding RNA transcriptomic profiling of mouse peritoneal macrophages spanning the whole lifespan of mice from 3-30 months. This analysis revealed an aged-dependent decrease in the expression of the micro RNA miR-146B. The authors further revealed that transient knock-down or knock-out of miR-146B expression in peritoneal macrophages leads to altered cytokine gene expression, indicative of skewed macrophage polarization seen in aging tissues. Additionally, miR-146B KO macrophages also had altered mitochondrial morphology and dysfunctional mitochondrial metabolism. Lastly, to further investigate peritoneal macrophage populations that are most affected by loss of miR-146B, the authors performed single-cell RNA sequencing of peritoneal macrophages from old and young WT and miR-146B KO mice. This analysis largely showed that gene expression in recruited monocyte derived (non-resident) macrophages is most affected by loss of miR-146B. The authors conclude that gradual loss of miR-146B may lead to macrophage dysfunction and inflammation phenotypes during aging.

Strengths:

– The authors use of unbiased non-coding RNA transcriptomic profiling of mouse peritoneal macrophages (spanning the whole lifespan of mice from 3-30 months) provides a thorough analysis of the expression profiles of multiple micro RNAs in an age-dependent manner. This approach allowed the authors to identify the age-dependent down-regulation in miR-146B expression. Furthermore, the authors data set also revealed age-dependent changes to other micro RNAs, which will be areas of future investigation and a great resource to the aging field.

– The authors utilized multiple genetics approaches to target miR-146B for transient knockdown and knockout in primary peritoneal macrophages and characterized the functional consequences. This approach led to the major findings in the paper that loss of miR-146B in macrophages leads to altered gene expression of inflammatory cytokines, metabolism genes, and mitochondrial dysfunction.

– In addition to in vitro based experiments, the authors developed a miR-146B KO mouse model and aged the mice to investigate how loss of miR-146B affected the aging phenotype of aging peritoneal macrophages. This analysis was largely done in an unbiased manner utilizing single-cell RNA sequencing.

Weakness:

– The authors claim that the unbiased non-coding RNA transcriptomic profiling revealed miR-146B as the only micro RNA with consistent progressive changes with age. However, careful analysis of Supplemental Figure 1A shows many other micro RNAs that appear to be both positively and negatively correlated with age, including miR-15a which has a nearly identical gene expression pattern to miR-146B.

– In Figure 1B, the authors attempt to validate the RNA-seq gene expression of miR-146B via qPCR in young and old peritoneal macrophages, to demonstrate that this micro RNA is down-regulated during aging. However, the authors do not attempt to look at other micro RNAs as controls to test their hypothesis that miR-146B is the only micro RNA whose expression is regulated during aging.

– Figure 1, the experimental details are not very clear as described in the main text or figure legends. For example, in all the experiments it is unclear whether the data represents individual mice or biological/technical replicates from an individual mouse, the sex of the mice used in the study is unclear, and many experiments have a small sample size, especially for being in vivo mouse experiments.

– The paper attempts to look at mechanisms of macrophage aging-related inflammation by solely focusing on thioglycolate induced peritoneal macrophages (a transient and non-resident monocyte derived subpopulation of macrophages) responding to acute inflammation. However, emerging evidence suggest that aging is characterized low-grade chronic inflammation. Thus, it's unclear whether miR-146B is relevant in aging-related inflammation since macrophages from naturally low-grade chronically inflamed aged tissues were not analyzed in this paper.

Furthermore, it is also unclear if miR-146B is differentially expressed in non-resident vs tissue resident macrophages as no attempt to measure this was done in the manuscript or in the single-cell data presented in Figure 4. This is especially relevant since the authors showed in Figure 1D that bone marrow derived macrophages do not express significant amounts of miR-146B, compared to peritoneal macrophages, suggesting that different population of macrophages may or may not express miR-146B, particularly those that drive inflammaging. Perhaps the authors could investigate macrophage phenotypes from other tissues (known to undergo inflammaging) such as fat tissue from young and old, WT and miR-146B KO mice.

– In Figure 2, the authors mention that loss of miR-146B affects macrophage polarization, skewing macrophages to a phenotype that resembles inflammaging. However, the authors only looked at a very narrow panel of cytokines. The data (both in vitro and in vivo) shows that loss of miR-146B leads to many pro-inflammatory cytokines being significantly down-regulated such as IL-1b and IL-6, while seeing an upregulation of the anti-inflammatory cytokine IL-10 suggesting miR-146B promotes an anti-inflammatory skewing (opposite of what the authors claim). Furthermore, this gene expression was performed under basal conditions which leads to less reliable gene expression. The authors did not attempt to measure how loss of miR-146B affects proper macrophage polarization via the treatment of macrophages with the type II cytokine IL-4 (M2) and LPS to skew to the classical M1 state.

– The data showing miR-146B regulates macrophage gene expression and mitochondrial function is descriptive and the authors do not provide any mechanistic insight in to how miR-146B promotes these changes.

-In Figure 4, the authors suggest the gene Lyz1 may be involved in the phenotype observed in miR-146B KO macrophages, but once again no attempt to demonstrate that miR-146B regulates mitochondrial function or gene expression via regulation of Lyz1 was performed. In fact, this conclusion is weakened by Figure 4E, showing that despite downregulation of miR-146B in old WT macrophages, Lyz1 expression does not increase as expected, making it an unlikely regulator of the altered gene expression seen in the WT old macrophages.

Overall, the authors of this study provided strong evidence that aging leads to the down-regulation of miR-146B expression in aging peritoneal monocyte derived macrophages. The authors have also provided strong evidence that miR-146B regulates cytokine expression and mitochondrial function in peritoneal macrophages. However, the paper suffers from being descriptive and lacking mechanistic insight in how miR-146B regulates macrophage cytokine expression and mitochondrial function. Lastly, for the reasons listed above the paper does not fully support the hypothesis that miR-146B may be a major driver of aging-related macrophage dysfunction and inflammaging.

1) Figure 1A, please clarify the units on the Y axis.

2) Typo pg 5, line 86, add space between number and months.

3) Figure 4D, clearly label what are resident vs non-resident markers.

4) Figure 4A and 4E, please list the genes in the same order and provide the same genes in each experiment.

5) Figure 4F, please label on graph what each Pattern represents, clearly state genes in each Pattern and if possible, show data for each gene in supplemental space.

[Editors' note: further revisions were suggested prior to acceptance, as described below.]

Thank you for resubmitting your work entitled "Loss of Mir146b with aging contributes to inflammation and mitochondrial dysfunction in thioglycollate-elicited peritoneal macrophages" for further consideration by *eLife*. Your revised article has been evaluated by Matt Kaeberlein (Senior Editor) and a Reviewing Editor.

The reviewers have discussed your revised submission, and found that crucial issues had not been addressed, as outlined below:

1. The purity panel needs to be more than n = 1 per age, and should be included in the manuscript, not just in the rebuttal letter. All reviewers were disappointed that this major point was not satisfactorily addressed.

2. In general, the authors should address all previous concerns raised in the first round of reviews that were not addressed, including:

– a number of the methodological points we raised (for instance the mix and match approach on genome reference usage mm9/mm10) are not at all addressed, not even textually in the revised manuscript.

– regarding the uniqueness of the miR-146B pattern, reviewers are not convinced. For instance, the authors do not attempt to look at other micro RNAs as controls to test their hypothesis that miR-146B is the only micro RNA whose expression is regulated during aging.

– information about biological vs. technical replication is still lacking in the revised manuscript.

---

## [Author Response]

Essential revisions:1) The authors use a type of "elicited" macrophages, Thioglycollate-elicited macrophages (TGEMs), which do not represent a naive state, but an activated/recruited state. However, this information is only included in the material and methods, and not discussed in the rest of the manuscript. Since this could have a great impact on the results, the three reviewers agreed that it is essential that authors explicitly address the use of TGEMs, (even if only textually), including in the title, abstract and main text, to make sure that the narrower scope of findings is clear to readers without needing to read the material and methods. If possible (maybe for future studies), similar analyses of other macrophage types would help understand the general relevance of the finding on the impact of miR-146b on inflamm-aging.

Use of Thioglycollate-elicited macrophages (TGEMs) has been has been explicitly highlighted throughout the manuscript, and we have included this detail in the revised manuscript title.

2) Since the authors opted for an adherence-only method of purification for the TGEMs, it is crucial that some measurement of the purity of the macrophage population be provided to make sure that the purity of TGEMs by adherence is not affected by aging. An F4/80 and Cd11b flow cytometry staining on cells purified similarly from the same ages and sexes would be the ideal method for this.

In our previous studies, we have established that aging does not affect the purity of the TGEM population selected by adherence (Lin JB, Sene A, Santeford A, et al. Oxysterol Signatures Distinguish Age-Related Macular Degeneration from Physiologic Aging. *EBioMedicine*. 2018;32:9-20. doi:10.1016/j.ebiom.2018.05.035). Using a flow cytometry approach, as was also suggested by reviewers here, we harvested TGEMs from female C57Bl/6 mice at approximately 3 and 18 months of age and stained for macrophage markers F4/80 (clone BM8) and CD64 (clone x54-5/7.1). We acquired data on a BD LSR II flow cytometer and used FlowJo v10 software to visualize and analyze data. No difference in the macrophage population was noted with either marker, and the overall percentages of cells positive for both macrophage markers (and therefore identified as macrophages) was consistently 94-96% in both old and young. Author response image 1 contains representative dot plots from TGEM samples from a young and an old mouse, each gated for live single cells and double positive for F4/80 and CD64, as well as normalized univariate histogram displays for each marker.

3) The authors need to carefully edit the manuscript to include all relevant and necessary methodological details (e.g. sex of used mice in general and by panel, a systematic clarification of the use of technical vs. biological replicates, etc.). The authors also should refer to the individual reviewer comments for the points needed clarification in the revised manuscript on this point.

Methodological details, including clarification of mouse sexes and ages and use of biological vs. technical replicates, have been extensively added throughout the manuscript for each experiment. This includes the Methods section as well as main text and figure legends.

4) Generally, the authors need to improve and amend their statistical analyses. This includes (i) providing more information on the analysis leading to only miR-146b (since referees note that other miRs look significant in the analysis), (ii) removal of all t-tests since there is no power to test for data normality, removal of statistical tests when the authors only have n = 2, etc.

We have added additional rationale regarding our identification of and focus on *miR-146b*, and acknowledged that other miRs were identified whose expression may change with mouse age and possibly warrant future investigation. In particular, reviewers noted that *miR-15a* levels represented in heatmap format in Supplemental Figure 1A look similar to those of *miR-146b*. Indeed, in this graphical representation pattern coloring does appear similar, particularly in small format. However, when we look directly at the numerical expression data, we can see that the decrease in expression from 3 months to 30 months in not unidirectional, as was seen with *miR-146b* and that may be expected as a result of the natural aging process. In fact, *miR-15a* expression actually increases by more than 10% between two separate consecutive time points (12 months to 18 months and 24 months to 30 months, respectively), thereby failing our criterion threshold.

**Author response image 2. respfig2:** 

We have also presented experiments throughout the manuscript that include a greater number of replicates, when possible, and amended our statistical analyses of all experiments in accordance with the recommendation to remove all t-test. For instances when comparison between two groups is necessary, we have utilized the non-parametric Mann-Whitney U-test. In addition, for discussion of cytokine gene expression in Figures 2C and F, we removed statistical analysis and referred only to trends in the data, while also providing additional data points from independent experiments.

5) Finally, potentially contradictory findings between figures need to be reconciled or explicitly discussed by the authors. (e.g. Reviewer #2 point 4)

We have amended the text to address the contradictory findings noted by the reviewers. Namely, we have addressed the discrepancy between in vitro knockdown and in vivo knockout of *miR-146b* in regards to cytokine gene expression levels. One potential explain of the differential patterns that we observed may be caused by the dramatic long-term (life-long) absence of *miR-146b* in conditional knockout macrophages vs. the short-term, partial reduction achieved through in vitro transfection.

We have also expanded our discussion of *Lyz1* data obtained through bulk RNA-seq and scRNA-seq. *Lyz1* was found to be the only gene significantly increasing as a result of *miR-146b* deletion in TGEMs in our bulk RNAseq analysis. Deeper analysis using scRNA-seq also found this target to be increased across all clusters between *miR-146b* and littermate controls at both young (3 months old) and old (17 months old) time points. However, as noted by Reviewer #2, we did not observe an increase in *Lyz1* when comparing old control TGEMs to young controls. As we have established that *miR-146b* expression is decreased with age in TGEMs, one may anticipate that expression of *Lyz1* should thereby increase. An important consideration is that the natural aging process leads to a slow and steady decline of *miR-146b*, though not a full obliteration of expression, whereas TGEMs from our conditional knockout mouse model show a persistent, near complete *miR-146b* loss. The continued expression of *miR-146b*, though lesser with age, in control/wildtype TGEMs may either be enough to continue regulating *Lyz1* and/or the slow decline in *miR146b* with aging may allow for additional, indirect compensatory regulation through other targets. These data illustrate in our opinion an important point about macrophage aging.

Reviewer #1:1) Methodological details need to be included or revised for consistency reproducibility.a. Please include a table with the sequences of all used qPCR and genotyping primers.

The list of qPCR probe sets and manufacturer ID/catalogue number has been presented in lieu of specific sequences. Each of the products used within this manuscript are commercially available. The manufacturers from which we obtained Taqman probes (Life Technologies) or LNA primers (Qiagen) do not disclose specific product sequences. However, utilizing the provided manufacturer assay IDs will allow other investigators to easily find these products should they choose to replicate these studies in their own hands.

All genotyping PCR sequences are presented in Supplemental Table 2 and the Key Resources table.

b. Some analyses are performed on the mm9 mouse genome build (e.g. small RNA-seq seq, line 449) and some on the mm10 genome build (e.g. RNA-seq of KO TGEMs line 591). Since this could lead to differences in results, please harmonize analyses so they are all performed on the same genomic build.c. Please include all code/scripts used for the analysis in a supplementary document or deposit them to a Github repository as per the journal policy.

References to GitHub depository have been moved to the Data Availability section for greater visibility and included in the Key Resources table.

Reviewer #2:1. The cytokine panel in Figure 2C (miR-146b knockdown) does not include some cytokines in 2F (miR-146b knockout) – were they measured? Also, please comment on the Mmp9 expression, which is decreased in 2C but increased in 2F.

In adding additional samples to our analyses, we have amended the list of cytokines to include targets measured for both knockdown and knockout, including some targets that may not be significantly changed. We have discussed any discrepancies within the text and noted possible rationale for differences noted in transgenic knockout vs. in vitro knockdown scenarios.

2. It would strengthen the paper if the levels of secreted cytokines (proteins) upon loss of miR-146b were measured.

We agree that protein levels often provide interesting insight to a cell’s biology. Our previous studies have shown that gene expression data provides a strong picture of the cell’s status. As our experiments did not utilize additional activators, such as *INFγ* + LPS or *IL4*, the greater sensitivity of qPCR over protein assays like ELISAs is capable of more accurately measuring differences here at baseline that may be otherwise less-reliably detectable due to the experimental limits of protein measuring.

3. In figure 3, the authors overexpress miR-146b and show increased mitochondrial respiration. Does this also alter the expression of cytokines measured in Figure 2?

The effects of *miR-146b* overexpression on TGEM cytokine or mitochondria-related gene expression have not yet been examined. These are interesting questions which certainly warrant follow-up studies, with both in vitro and possibly mouse models.

4. The miR-146b-dependent metabolic shift may result from alterations of multiple metabolic pathways that consequently affect OCR/ECAR, such as glucose metabolism. Were there metabolic genes that changed in the RNA-seq? If so, is/were there a coherent metabolic pathway(s) that is/are highlighted? If possible, quantifying metabolites that are highly relevant to macrophage function would provide further insight.

RNA-seq revealed statistically significant decreases in several genes involved in mitochondrial morphology and respiration, as noted in lines 342-3. However, no one pathway in particular seems to be targeted, though pathway analysis without a large number of genes can be limiting. Metabolomics analysis is great idea for future follow up studies, but is beyond the scope of the current project.

5. Are the cytokines measured in Figure 2 reflected in the scRNA-seq of Lyz2; miR-146bM-/M- mice?

Most of the cytokines measured in Figure 2 do appear in the scRNA-seq data set, as shown in Author response image 3. As many of the reads are scant, however, it is difficult to make broad assumptions regarding these targets in this context. For this reason, we did not include the data within the manuscript.

**Author response image 3. respfig3:** 

6. As the authors state, peritoneal macrophages consist of a heterogeneous population of resident and recruited (monocyte-derived) macrophages. Further, monocyte-derived macrophages may not display age-dependent loss of miR-146b (Figure 1C). The authors may want to add some discussion on the potentially differential role of resident vs recruited macrophages in inflammaging. Further, have the authors tried to compare resident vs recruited macrophages in the scRNA-seq on peritoneal macrophages in addition to the 3 clusters (it is not clear whether the largest cluster 1 is a mix of both populations)?

BMDMs indeed exhibit age-related loss of *miR-146b*, though perhaps not as dramatically as noted in TGEMs. The levels of *miR-146b* even in BMDMs from young mice however is only ~30% that of TGEMs from age-matched mice. In the scRNA-seq, the majority of cells expressing markers or resident peritoneal macs map to cluster 2. However, the vast majority of cells map to cluster 1, as noted in Supplemental B, and as such would be anticipated to contribute most to the phenotypes we observed.

Reviewer #3:1) Figure 1A, please clarify the units on the Y axis.

Y axis units of Figure 1A have been amended to “Normalized Expression (RPM)”. Units represent the normalized expression value of reads per million mapped to the mouse genome.

2) Typo pg 5, line 86, add space between number and months.

Spacing between number and months has been added to p5 line 86.

3) Figure 4D, clearly label what are resident vs non-resident markers.

Labels have been added to Figure 4D indicating resident vs recruited markers.

4) Figure 4A and 4E, please list the genes in the same order and provide the same genes in each experiment.

Gene listings for Figures 4A and 4E have been harmonized to list genes in the same order.

5) Figure 4F, please label on graph what each Pattern represents, clearly state genes in each Pattern and if possible show data for each gene in supplemental space.

For Figure 4F, the main text and figure legend has been expanded to more clearly explain that each row represents the mean z score for that hierarchically defined cluster of genes from the heatmap in Supplemental Figure 3. This was used to convey the various expression patterns across our four samples, and to interpret the GO categories and functions of the groups of genes with these patterns. All genes from Patterns a-e are noted in Supplemental Figure 3, along with their individual expression patterns across each sample, as well as listed Supplemental Table 1 for additional clarity.

[Editors' note: further revisions were suggested prior to acceptance, as described below.]

The reviewers have discussed your revised submission, and found that crucial issues had not been addressed, as outlined below:1. The purity panel needs to be more than n = 1 per age, and should be included in the manuscript, not just in the rebuttal letter. All reviewers were disappointed that this major point was not satisfactorily addressed.2. In general, the authors should address all previous concerns raised in the first round of reviews that were not addressed, including:– a number of the methodological points we raised (for instance the mix and match approach on genome reference usage mm9/mm10) are not at all addressed, not even textually in the revised manuscript.– regarding the uniqueness of the miR-146B pattern, reviewers are not convinced. For instance, the authors do not attempt to look at other micro RNAs as controls to test their hypothesis that miR-146B is the only micro RNA whose expression is regulated during aging.– information about biological vs. technical replication is still lacking in the revised manuscript.

1. We have included flow cytometry analysis of both young (3 month old, n=5 biological replicates/individual mice) and old (20+ month old, n=5 biological replicates/individual mice) TGEMs using the macrophage markers CD11b and F4/80 to validate the purity of the macrophage population utilizing the adherence selection method employed throughout this study. Approximately 95% of all cells, in both young and old samples, were double positive for both markers, indicating not only a high level of macrophage purity, but also demonstrating that there is no purity difference due to age. This data has been included as Figure 1—figure supplement 1 and Figure 1—figure supplement 1 source data. Dr. Lynn Hassman contributed to this effort and has been included as an author on the revised manuscript.

2. We have evaluated the miRNA expression patterns for a number of additional miRNAs in TGEMs. Here we further investigated microRNAs identified in our original dataset to have overall decreased expression with age, but which were not noted to decline in quite the linear fashion with each age point, we observed *Mir146b*. Based on values from our original small RNA-seq, we have evaluated the expression of *Mir15a, Mir22, Mir423, Mir29a, Mir146a,* and *Mir18a* along with *Mir146b*. We also attempted to analyze *Mir362,* but found its expression below the limit of detection for nearly half of our samples, regardless of age, and therefore did not include it in this manuscript. While our original experiment was able to utilize mice as old as 30 months of age, 20 month old mice is the oldest time point that we are able to presently acquire. We have noted this caveat within the manuscript. This new validation using mice using 7-9 individual mice at 3, 12, and 20 months of age again demonstrated decline in *Mir146b* from 3 to 20 months. In addition, *Mir22* was observed to decline between these time points as well. Our RNA-seq data indicates that while there may be decreases in expression between 3 and 18-24 months, expression may actually increase again from 24-30 months. We have included the graph of normalized expression values for *Mir22* obtained from our small RNA-seq data, but we cannot presently validate this potential increase in TGEMs from 30 month mice, and as such, do not comment on this pattern within the manuscript. We do however note that the level of reads for *Mir146b* is more than 3 fold higher than that of *Mir22,* keeping *Mir146b* an attractive target for our study, but also note the need for future studies of *Mir22* in the aging macrophage.

3. We have explicitly highlighted and discussed that our original small RNA-seq data, which helped us to identify *Mir146b* as a microRNA of interest, was aligned to the mm9 version of the mouse genome. We understand that re-analysis of the data using a newer version may reveal different patterns of expression for some miRNAs, revealing additional miRNAs of interesting in aging. However, due to the changes in raw data file types and programs that have occurred in the time period since this data was originally procured, conversion has been technically challenging at this point, and as such we are currently unable to realign these files. Importantly though, our qPCR of miRNA expression and additional experiments validates our initial findings from this RNAseq data set—*Mir146b* progressively declines with aging in murine TGEMs.

4. We have reviewed details in both the manuscript body as well as figure legends to ensure that we have explicitly indicated the use of biological vs technical replicates of each experiment and added additional methodological details within both the results and Materials and methods sections as well as in the figure legends.

5. Use of Thioglycollate-elicited macrophages (TGEMs) has been has been explicitly highlighted throughout the manuscript, and we have included this detail in the revised manuscript title.

6. We have presented experiments throughout the manuscript that include a greater number of replicates, when possible, and amended our statistical analyses of all experiments in accordance with the recommendation to remove all t-test. For instances when comparison between two groups is necessary, we have utilized the non-parametric Mann-Whitney U-test. In addition, for discussion of cytokine gene expression in Figures 2C and F, we removed statistical analysis and referred only to trends in the data, while also providing additional data points from independent experiments.

7. We have amended the text to address the contradictory findings noted by the reviewers. Namely, we have addressed the discrepancy between in vitro knockdown and in vivo knockout of *miR-146b* in regards to cytokine gene expression levels. One potential explain of the differential patterns that we observed may be caused by the dramatic long-term (life-long) absence of *miR-146b* in conditional knockout macrophages vs. the short-term, partial reduction achieved through in vitro transfection.

We have also expanded our discussion of *Lyz1* data obtained through bulk RNA-seq and scRNA-seq. *Lyz1* was found to be the only gene significantly increasing as a result of *miR-146b* deletion in TGEMs in our bulk RNAseq analysis. Deeper analysis using scRNA-seq also found this target to be increased across all clusters between *miR-146b* and littermate controls at both young (3 months old) and old (17 months old) time points. However, as noted by Reviewer #2, we did not observe an increase in *Lyz1* when comparing old control TGEMs to young controls. As we have established that *miR-146b* expression is decreased with age in TGEMs, one may anticipate that expression of *Lyz1* should thereby increase. An important consideration is that the natural aging process leads to a slow and steady decline of *miR-146b*, though not a full obliteration of expression, whereas TGEMs from our conditional knockout mouse model show a persistent, near complete *miR-146b* loss. The continued expression of *miR-146b*, though lesser with age, in control/wildtype TGEMs may either be enough to continue regulating *Lyz1* and/or the slow decline in *miR146b* with aging may allow for additional, indirect compensatory regulation through other targets. These data illustrate in our opinion an important point about macrophage aging.